

# On the model uncertainties in Bayesian source reconstruction using the emission inverse modelling system FREARtool v1.0 and the Lagrangian transport and dispersion model Flexpart v9.0.2

Pieter De Meutter[1,2,3], Ian Hoffman[1], and Kurt Ungar[1]

[1]Radiation Protection Bureau, Health Canada, 775 Brookfield Road, Ottawa, Canada
[2]Belgian Nuclear Research Institute, Boeretang 200, Mol, Belgium
[3]Royal Meteorological Institute of Belgium, Ringlaan 3, Brussels, Belgium

**Correspondence:** Pieter De Meutter (pieter.de.meutter@sckcen.be)

**Abstract.** Bayesian source reconstruction is a powerful tool for determining atmospheric releases. It can be used, amongst other applications, to identify a point source releasing radioactive particles into the atmosphere. This is relevant for applications such as emergency response in case of a nuclear accident, or Comprehensive Nuclear-Test-Ban treaty verification. The method involves solving an inverse problem using environmental radioactivity observations and atmospheric transport models.

The Bayesian approach has the advantage of providing credible intervals on the inferred source parameters in a natural way. However, it requires the specification of the inference input errors, such as the observation error and model error. The latter is particularly hard to provide as there is no straightforward way to determine the atmospheric transport and dispersion model error. Here, the importance of model error is illustrated for Bayesian source reconstruction using a recent and unique case where radionuclides were detected on several continents. A numerical weather prediction ensemble is used to create an ensemble of

atmospheric transport and dispersion simulations, and a method is proposed to determine the model error.

## 1 Introduction

Nuclear facilities release a certain amount of anthropogenic radioactive particulates or gases into the atmosphere, which are transported and dispersed by the wind. These releases can either be routine or accidental. Several countries run a network of

stations to monitor airborne levels of environmental radioactivity (Steinhauser, 2018). These monitoring networks allow to verify compliance with regulatory release limits, but also to detect reported and unreported nuclear events. Recent examples include the detections of [131]I in Europe (Masson et al., 2018) or [106]Ru detections on the northern hemisphere (Masson et al., 2019).

On the international scale, the radionuclide component of the International Monitoring System will consist of 80 stations

measuring radioactive particulates (of which at least 40 will be equipped with radioactive noble gas detectors). This network





is being set up to verify compliance with the Comprehensive Nuclear-Test-Ban Treaty once it enters into force. In the past, anomalous radionuclide detections were made that are likely linked to a nuclear explosion (Ringbom et al., 2014; De Meutter et al., 2018).

If an anomalous detection occurred (either from a nuclear accident or a clandestine nuclear weapon test), methods are needed that relate the detection with its source - or potential sources if the source is unknown. One of these methods is atmospheric transport and dispersion modelling. An atmospheric transport and dispersion model typically simulates the transport, dispersion, dry and wet deposition and radioactive decay of radionuclides released in the atmosphere. These processes establish a linear relationship between the concentrations at receptors and the release amount at the source. One can calculate such source–receptor-relationships (Seibert and Frank, 2004) between a fixed source and several receptors or stations (when modelling forward in time) or between several potential sources and a fixed receptor (when modelling backwards in time).

A significant event of interest will often be accompanied by multiple detections taken at multiple stations. Statistical methods can then be employed to combine the information from all these detections (and possibly non-detections) in a meaningful way in order to infer relevant information on the source. In case of an unknown source, the objective is often to find the source location, release time and release amount. In case of a known source, the source location and perhaps also the release times are known. In that case, the release amount and release height can be inferred to refine a previous release estimate obtained through other ways (for instance, an estimation could be made based on accident scenarios and the known or estimated inventory of a reactor). The process of inferring information on the source based on observations is called inverse modelling. Several methods exist, ranging from simply calculating correlations between observations and source-receptor-sensitivities to locate the source (Becker et al., 2007) to more elaborate methods such as optimization methods using cost functions (e.g., Stohl et al., 2012), data assimilation (e.g., Bocquet, 2007) or Bayesian inference (e.g., Yee, 2012).

Of these methods, the Bayesian inference has the advantage of readily providing an uncertainty quantification on the outcome. However, the quality of the inference and the uncertainty quantification depends on the quality of input uncertainties. Typically, one specifies the observation error and model error. Here, the model error relates to errors in the atmospheric transport and dispersion model. These errors are very hard to readily quantify, mainly because of the underlying numerical weather prediction data which is used to calculate the transport and dispersion. The errors associated with numerical weather prediction depend on the atmospheric state and thus vary between locations and from day to day.

A well-established method to quantify uncertainties in numerical weather predictions is the ensemble method (Leutbecher and Palmer, 2008). For simplicity, let us focus on single-model ensembles. Such ensembles are created using a single model and consist of a set of one unperturbed and several perturbed scenarios or model predictions. These scenarios are designed in such a way that the spread among the different scenarios represent the uncertainty of an individual model prediction. Each scenario (also called an ensemble member) is created by perturbing certain input data and/or using different parameterisation schemes (except for the unperturbed member, which is created by running the model with the best available input and parameterisation schemes). The key to create a good ensemble lies in providing realistic perturbations. Ideally, one has a large number of distinct ensemble members (so that many different but realistic scenarios can be obtained). However, the huge computational cost of running an ensemble and storing its vast amount of data limits the number of ensemble members. Therefore, ensembles used





operationally at major weather institutes around the world are designed in a way that, even with a limited number of members (between 14 and 50, Leutbecher, 2019), the ensemble can capture most of the possible outcomes.

Yee et al. (2014) demonstrated the significant impact of model error on the outcome of Bayesian source reconstruction by employing two different measurement models for the incorporation of the model error. Nevertheless, given the difficulty to
quantify the model error, many studies instead rely on assumptions regarding the model error structure and scale. In this paper, an ensemble of atmospheric transport model simulations is used to determine the model error used in the Bayesian inference. The effect of different model error formulations on the source localisation is studied for a recent case where $^{106}$Ru was observed throughout the northern hemisphere in autumn 2017. In this study, the Bayesian source reconstruction tool described in De Meutter and Hoffman (2020) will be used.

In Section 2, the case that is used to illustrate how the ensemble can be used to determine the model error, is described. The observational and model data are described. Section 3 describes the Bayesian inference system. Section 4 describes the effect of model error on the source location probability maps. The uncertainty parameters are fitted using the ensemble in Section 5. The ensemble is used in a scenario-based way in Section 6. This allows to test whether information is lost when using the ensemble solely to fit uncertainty parameters as performed in Section 5. Finally, conclusions are given in Section 7.

## 2 Description of the case

In autumn 2017, several national and international monitoring networks reported the detection of $^{106}$Ru and, to a lesser extent $^{103}$Ru (Masson et al., 2019). However, to date, no country or facility has claimed responsibility for the release. $^{106}$Ru (half-life: 373.6 d) and $^{103}$Ru (half-life: 39.26 d) are radioactive particulates that do not have natural sources. Since no other fission products such as iodine and cesium were measured, a nuclear reactor accident can be excluded. Several studies using direct
and inverse atmospheric transport modelling showed that a release in the region of the southern Ural mountains in Russia can best explain the observations (Sørensen, 2018; Saunier et al., 2019; Bossew et al., 2019; De Meutter et al., 2020). Two major nuclear facilities are located in that area: the Research Institute of Atomic Reactors and the Mayak Production Association (Fig. 1).

Here, we revisit the modelling data used in De Meutter et al. (2020) but perform a Bayesian analysis instead of a cost function
analysis. We apply our methods to this case as it offers a very interesting data set since (i) there were detections on multiple continents, (ii) the single-source assumption is likely valid as there is no measurable global background from anthropogenic sources and (iii) it is a recent case so that state-of-the-art numerical weather data is readily available. However, we stress that the case study is used here to examine the model error structure for a Bayesian source reconstruction, and how an ensemble can provide insight in the model error structure and scale – it is not our goal as such to find the origin of the reported $^{106}$Ru
and $^{103}$Ru.





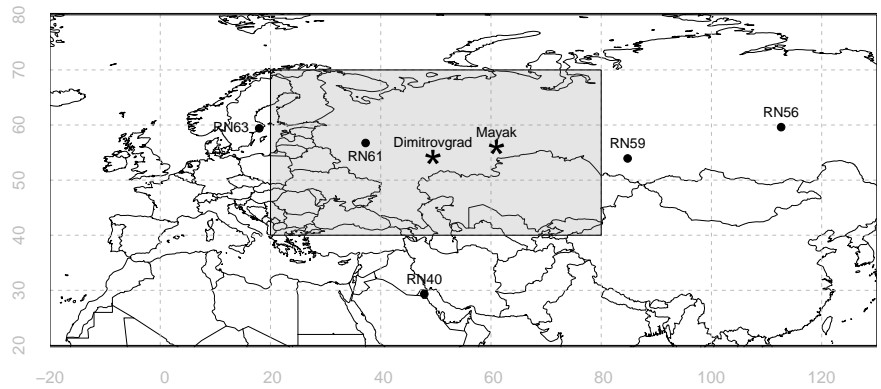

**Figure 1.** Locations of the five stations from which Ru-106 detections have been used. The shaded box shows the search domain of the Bayesian inference. For reference, the locations of two nuclear facilities are also shown: the Research Institute of Atomic Reactors in Dimitrovgrad, labelled "Dimitrovgrad" (54.19° lat and 49.48° lon) and the Mayak Production Association in Ozyorsk, labelled "Mayak" (55.70° lat and 60.80° lon).

## 2.1 Activity concentration observations

Twelve $^{106}$Ru detections have been used from five different stations of the International Monitoring System which is being commissioned to verify compliance with the Comprehensive Nuclear-Test-Ban Treaty once it enters into force: three consecutive detections at station RN40 (Kuwait City, Kuwait) taken between 4 and 6 October 2017; one detection at station RN56 (Peleduy, Russian Federation) taken on 6 October 2017 ; four consecutive detections at station RN59 (Zalesovo, Russian Federation) taken between 3 and 6 October 2017; one detection at station RN61 (Dubna, Russian Federation) taken on 5 October 2017; and three consecutive detections at station RN63 (Stockholm, Sweden) taken between 1 and 3 October 2017. The location of these stations are shown in Fig. 1. Although a few stations also measured $^{103}$Ru, these detections have not been used for the inference. The stations sample approximately 20,000 m$^3$ of air during a period of 24 h, during which radioactive particulates - if present - are captured on a filter. The filter is then analyzed using gamma spectroscopy to detect the presence of any radionuclides. The above observation times refer to the collection stop of the sample. Although the observations were selected to realistically represent this case, we do not attempt to use all (or some optimal selection) of the available detections and non-detections in the inference as it is not our goal to determine the true source location as such.

## 2.2 NWP and atmospheric transport modelling

We have used the source–receptor-sensitivities associated with the twelve observations from De Meutter et al. (2020). These were obtained by running the Lagrangian particle model Flexpart (Stohl et al., 2005) in backward mode (Seibert and Frank, 2004). The source–receptor-sensitivities represent the residence time of modelled particles in a geotemporal grid box. The model output frequency was three hours, so that the maximum possible residence time in a geotemporal grid box is 10,800 s.





The source–receptor-sensitivities have horizontal grid spacings of 0.5°. An activity concentration (in Bq/m$^3$) can be related to a
release (in Bq) by multiplying the latter with the source–receptor-sensitivity (in s), and dividing by the grid box volume and the
output frequency. An ensemble of numerical weather predictions was used to create an ensemble of atmospheric transport and
dispersion simulations. The transport and dispersion processes themselves were not perturbed. The Ensemble of Data Assimi-
lations from the European Centre for Medium-Range Weather Forecasts (ECMWF) was used to run Flexpart. It consists of 26
independent lower-resolution 4D-Var assimilations: one using unperturbed observations and physics, and 25 using perturbed
observations, sea-surface temperatures and model physics (Bonavita et al., 2016). By adding and subtracting perturbations from
the ensemble mean, the number of perturbed members was doubled, so that 50 perturbed and one unperturbed member were
obtained. The perturbations are created in such a way that each ensemble member represents a possible scenario for the true
atmospheric state, and the spread between the different members represents the uncertainty. Then, for each weather ensemble
member, Flexpart was run so that an atmospheric transport ensemble of 51 source-receptor-sensitivities was obtained.

## 3 Bayesian source reconstruction

De Meutter and Hoffman (2020) have developed and applied a Bayesian inference system to find the source parameters of
an anomalous $^{75}$Se release based on airborne $^{75}$Se activity concentrations. The main components of the inference system are
summarized here, but details are given in De Meutter and Hoffman (2020) and references therein.

### 3.1 Source parameters

The unknown source is described by five source parameters:

- the longitude of the source ($x_s$)

- the latitude of the source ($y_s$)

- the accumulated release ($Q$)

- the release start time ($t_{start}$)

- the release end time ($t_{stop}$)

In practice, the release period is parameterized by fractions $rstart$ and $rstop$ to ensure that $t_{start}$ occurs before $t_{stop}$. The
corresponding release period is calculated as follows:

$$t_{start} = t_1 + rstart \left( t_m - t_1 \right) \tag{1}$$

$$t_{stop} = t_{start} + rstop \left( t_m - t_{start} \right) \tag{2}$$





with $t_1$ and $t_m$ the first and last time for which source–receptor-sensitivities are available for the source reconstruction. The
release rate is assumed constant during the release period. The vertical position of the source is assumed to stretch between the
surface and the top of the lowest model layer (at 100 m). Thus, the unknown source is parameterized as follows:

$$\Xi(x,y,z,t;x_s,y_s,z_{top},Q,t_{start},t_{stop}) = \frac{Q}{(t_{stop}-t_{start})\,z_{top}}\,\delta(x-x_s)\,\delta(y-y_s)\,[\mathcal{H}(z)-\mathcal{H}(z-z_{top})]$$

$$[\mathcal{H}(t-t_{start})-\mathcal{H}(t-t_{stop})] \tag{3}$$

with $\delta$ the Dirac delta function and $\mathcal{H}$ the Heaviside step function.

## 3.2   Prior distribution

Uninformative bounded uniform priors are used for the source parameters. The source longitude is assumed to be between
$20°$ and $80°$ and the source latitude is assumed to be between $40°$ and $70°$ (see Fig. 1 for a map showing the search domain).
The accumulated release is assumed to be between $10^{10}$ Bq and $10^{16}$ Bq. Since this spans many orders of magnitude, we take
$\log 10(Q)$ as source parameter in our implementation and simply impose a uniform prior between 10 and 16. Recall that the
first observation that we consider for the inference was taken on 1 October 2017. Therefore, the release is assumed to have
occurred between 25 September 2017 0000 UTC and 28 September 2017 0000 UTC. Generally, the upper limit on the release
time will exclude solutions further downwind, while the lower limit on the release time will exclude solutions further upwind.
Uniform bounded priors between 0 and 1 are used for $rstart$ and $rstop$.

## 3.3   Likelihood

De Meutter and Hoffman (2020) proposed likelihood equations that can take into account detections, instrumental non-
detections, misses and false alarms using Currie detection limits (Currie, 1968). Since non-detections will not be used in
this study, only the likelihood of detections will be used here. The possibility of a false alarm, where the detector wrongly
identifies a detection, is also considered. For simplicity, the observations $\mathbf{c_{det}}$ are assumed to be independent, thereby neglect-
ing possible geotemporal correlations. As a result, the total likelihood is simply the product of the likelihood associated with
individual observations:

$$p(\mathbf{c_{det}}|\mathbf{c_{mod}}) = \prod_{i=1}^{n} p(c_{det,i}|c_{mod,i}) \tag{4}$$

First, let us define $c_{true}$, which is the true activity concentration that will never be known. Next, we define $c_{det}$, which is the
activity concentration as seen by the detector and which can differ from $c_{true}$ (the observed net signal can even be negative
due to the statistical nature of spectroscopic analysis). We use the Currie critical threshold, $L_C$, (Currie, 1968) to decide, given
a net signal, whether a real detection took place ($d_{obs}$ if $c_{det} > L_C$) or not ($\bar{d}_{obs}$ if $c_{det} < L_C$). One can then define a true
non-detection, a miss, a true detection and a false alarm as in Table 1. In this application, the risk of a missed detection and





**Table 1.** Definition of a true non-detection, a miss, a false alarm and a true detection based on the Currie critical level $L_C$, the detected activity concentration $c_{det}$ and the "true" activity concentration $c_{true}$ which will never be known.

|  | $\bar{d}_{obs}\,(c_{det} < L_C)$ | $d_{obs}\,(c_{det} > L_C)$ |
|---|---|---|
| $\bar{d}_{true}\,(c_{true} < L_C)$ | true non-detection | false alarm |
| $d_{true}\,(c_{true} > L_C)$ | miss | true detection |

a false alarm is set equal to 5 %. $c_{true}$ will never be known, but we assume that the modelled activity concentration $c_{mod}$ corresponding to a source hypothesis $\Xi$ and its associated uncertainty results in a distribution of true activity concentrations $d_{c_{true}}$ given by the following formula:

$$d_{c_{true}}(c_{true}|c_{mod,i}) = \frac{\bar{\alpha}^{\bar{\beta}}\,\Gamma(\bar{\beta}+0.5)}{\sqrt{2\pi}\,s_i\,\Gamma(\bar{\beta})}\,\frac{1}{[\bar{\alpha}+(c_{true}-c_{mod,i})^2/(2\,s_i^2)]^{\bar{\beta}+0.5}} \tag{5}$$


with the index $i$ denoting "corresponding to the i$^{th}$ observation" (with $i = 1...12$), $\Gamma$ the gamma function and $s$, $\bar{\alpha}$ and $\bar{\beta}$ parameters of an inverse gamma distribution. The above equation was used by Yee (2012) as likelihood function for detections, and was obtained by starting from a Gaussian function, replacing the standard deviation $\sigma$ by an inverse gamma distribution and integrating over all possible values of $\sigma$. However, in order to take into account the possibility of a false alarm, instead we propose the following likelihood for the i$^{th}$ detection $c_{det,i}$ given its corresponding modelled activity concentration $c_{mod,i}$ associated with a source hypothesis $\Xi$:


$$p(c_{det,i}|c_{mod,i}) = p(c_{det,i}|c_{mod,i},d_{true})\,p(d_{true}|c_{mod,i}) + p(c_{det,i}|\bar{d}_{true})\,p(\bar{d}_{true}|c_{mod,i}). \tag{6}$$

In the equation above, $p(d_{true}|c_{mod})$ is the probability of a true detection, given by:

$$p(d_{true}|c_{mod}) = \int_0^\infty p(d_{true}|c_{true})\,p(c_{true}|c_{mod})\,\mathrm{d}c_{true} \tag{7}$$


$$= \int_{L_C}^\infty p(c_{true}|c_{mod})\,\mathrm{d}c_{true}, \tag{8}$$

and $p(\bar{d}_{true}|c_{mod})$ the probability of a true non-detection, given by:



$$p(\bar{d}_{true}|c_{mod}) = \int_0^\infty p(\bar{d}_{true}|c_{true})\,p(c_{true}|c_{mod})\,\mathrm{d}c_{true} \tag{9}$$

$$= \int_0^{L_C} p(c_{true}|c_{mod})\,\mathrm{d}c_{true} \tag{10}$$

$$= 1 - p(d_{true}|c_{mod}). \tag{11}$$

In these equations, $p(c_{true}|c_{mod})$ is simply equal to Eq. 5, but normalized (below, $c'_{true}$ is a dummy variable for integration):

$$p(c_{true}|c_{mod}) = \frac{d_{c_{true}}(c_{true}|c_{mod})}{\int_0^\infty d_{c_{true}}(c'_{true}|c_{mod})\,\mathrm{d}c'_{true}} \tag{12}$$

$p(c_{det,i}|c_{mod,i}, d_{true})$ gives the likelihood of detecting $c_{det,i}$ given $c_{mod,i}$ and assuming that the detection was not a false alarm. We assume it is given by Eq. 5 as follows:

$$p(c_{det,i}|c_{mod,i}, d_{true}) = d_{c_{true}}(c_{true} = c_{det,i}|c_{mod,i}) \tag{13}$$

$p(c_{det,i}|\bar{d}_{true})$ is the likelihood of detecting $c_{det,i}$ assuming that the detection is a false alarm:

$$p(c_{det,i}|\bar{d}_{true}) = \int_0^{L_C} \frac{1}{\sqrt{2\pi}L_C/k_\alpha} \exp\left(-\frac{(c_{det} - c_{true})^2}{(L_C/k_\alpha)^2}\right)\,\mathrm{d}c_{true} \tag{14}$$

with $k_\alpha = 1.645$ for a false alarm risk of 5 %. The likelihood and its different components are plotted for two hypothetical detections in Fig. 2. For the hypothetical detection $c_{det}$ close to its $L_C$ (Fig. 2, left), the likelihood is significantly increased for small $c_{mod}$ because the possibility of a false alarm is considered. For a hypothetical detection $c_{det}$ significantly larger than its $L_C$ (Fig. 2, right), the consideration of a false alarm does not significantly add to the total likelihood. Indeed, for $c_{det,i} \gg L_{C,i}$ and $c_{mod,i} \gg L_{C,i}$, Eq. 6 simplifies to:

$$p(c_{det,i}|c_{mod,i}) \approx d_{c_{true}}(c_{true} = c_{det,i}|c_{mod,i}) \tag{15}$$

### 3.4  Sampling from the posterior

The general-purpose Markov chain Monte Carlo algorithm MT-DREAM$_{(ZS)}$ (Laloy and Vrugt, 2012) is used to sample the posterior distribution. A chain of source hypotheses is obtained by repeatedly creating a proposal source hypothesis based on the current source hypothesis, and then accepting the proposal or - if rejected - retaining the current source hypothesis.





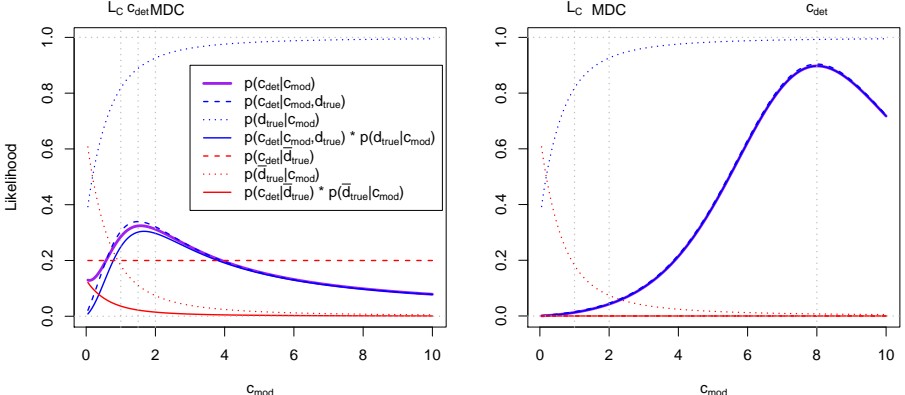

**Figure 2.** Likelihood function for two detections: $c_{det} = 1.5$ (left) and $c_{det} = 8.0$ (right). The gray dotted vertical lines represent $L_C = 1.0$, MDC $= 2.0$ and the detected activity concentration $c_{det}$. The units are arbitrary in this example.

A key feature of a Markov chain Monte Carlo algorithm is its ability to construct proposals in such a way that the posterior distribution is efficiently explored and sampled. MT-DREAM$_{(ZS)}$ creates a new proposal by adding a perturbation to the current source hypothesis. Such perturbations are created by taking the difference of two randomly drawn states out of an archive of past states, so that the size and direction of the perturbation are adaptive and optimal to the problem - without the necessity of

prior tuning. Every 10$^{th}$ iteration, the current source hypothesis is added to the archive. Three chains are run simultaneously (sharing the same archive of past states) to diagnose convergence more easily. The algorithm is designed so that a snooker step occurs with a probability of 20 % to allow jumps between different posterior modes (ter Braak and Vrugt, 2008). To enhance efficiency and to obtain more accurate results, randomized subspace sampling is used (Vrugt et al., 2009). Furthermore, MT-

DREAM$_{(ZS)}$ makes use of multiple try Metropolis sampling (Liu et al., 2000) to enhance the mixing of the chains.

### 3.5 Observation and model errors

The detected activity concentration $c_{det}$ is assumed to be Gaussian distributed around the true activity concentration $c_{true}$, with a standard deviation $\sigma_{obs} = L_C/k_\alpha$ as in Eq. 14. The model error originates from errors in the source-receptor-sensitivities calculated by the atmospheric transport model. The main sources of error are simplifications in the turbulence parameterization

of the atmospheric transport model and errors in the numerical weather prediction data used to run the atmospheric transport model. As it is very hard if not impossible to specify the model error, we assume Gaussian errors, but replace a fixed $\sigma_{mod}$ by an inverse gamma distribution, as mentioned earlier in this section and following Yee (2012):

$$\psi(\sigma_{mod,i}|s_i, \bar{\alpha}_i, \bar{\beta}_i) = 2\, \frac{\bar{\alpha}_i^{\bar{\beta}_i}}{\Gamma(\bar{\beta}_i)} \left( \frac{s_i}{\sigma_{mod,i}} \right)^{2\bar{\beta}_i} \exp\left( -\bar{\alpha}_i\, \frac{s_i^2}{\sigma_{mod,i}^2} \right) \frac{1}{\sigma_{mod,i}} \tag{16}$$

with the subscript $i$ denoting that these values can be observation-specific. The parameters $s$, $\bar{\alpha}$ and $\bar{\beta}$ of the inverse gamma

distribution are respectively an estimate of $\sigma$, a scale parameter and a shape parameter. Yee (2012) proposed to use $\bar{\alpha} = \pi^{-1}$





and $\bar{\beta} = 1$, in which case $\langle \sigma_{mod,i} \rangle = s_i$ and the variance of $\sigma_{mod,i}$ becomes infinite (Eq. 16 will then be a very heavy-tailed distribution). We now have to come up with a value for $s_i$. Since the source-receptor-sensitivities typically span many orders of magnitude, it makes more sense to define a relative error rather than an absolute error. Furthermore, because the source-receptor-sensitivities are linearly proportional to the activity concentration, we propose the following for $s_i$:

$$s_i = \frac{\sigma_{srs,i}}{srs_i} \max(c_{det,i}, 16 * L_{C,i}) \tag{17}$$

As a consequence, the model uncertainty does not depend on the source parameters. Note that the above proposal results in a larger relative model error if $c_{det,i} < 16 * L_{C,i}$. This is desirable since small detections are caused by a part of the plume of radionuclides that was subject to more atmospheric transport and dispersion processes and thus should have a larger relative uncertainty than large detections (although the latter have higher absolute uncertainty). In Eq. 13, $s_i$ is replaced by the following in order to take into account both observation and model error:

$$s_i \to \sqrt{s_i^2 + \sigma_{obs,i}^2} \tag{18}$$

From now on, we will express any parameters $s_i$ and $\sigma$ as relative errors, thus as fractions of $\max(c_{det,i}, 16 * L_{C,i})$ (see Eq. 17).

## 4 Model error and Bayesian source reconstruction

### 4.1 Posterior effects

In this section, the effect of model error on the posterior is illustrated by varying the parameter $s_i$ of the inverse gamma function (Eq. 16), which roughly fixes the scale of the model uncertainty. The Bayesian inference was independently repeated multiple times, once for each of the chosen parameter values. The resulting source location probability maps are shown in Fig. 3 for three values of $s_i$: 0.3, 0.5 and 3. Here, the same value $s_i$ is used for all observations. The source location probability maps illustrate that the model error has a profound impact on the resulting posterior. The source location probability ranges from a narrow spot to a fairly elongated area. Perhaps surprisingly, an increase in the uncertainty parameter $s$ does not simply enlarge the region of possible sources: instead, the enlargement is mostly in one direction (northeast); furthermore, the area with the highest probability shifts (also northeast). Therefore, one cannot simply predict beforehand how the posterior will change when the model error is changed. The resulting source location is in line with what previous studies found (Sørensen, 2018; Saunier et al., 2019; Bossew et al., 2019; De Meutter et al., 2020).



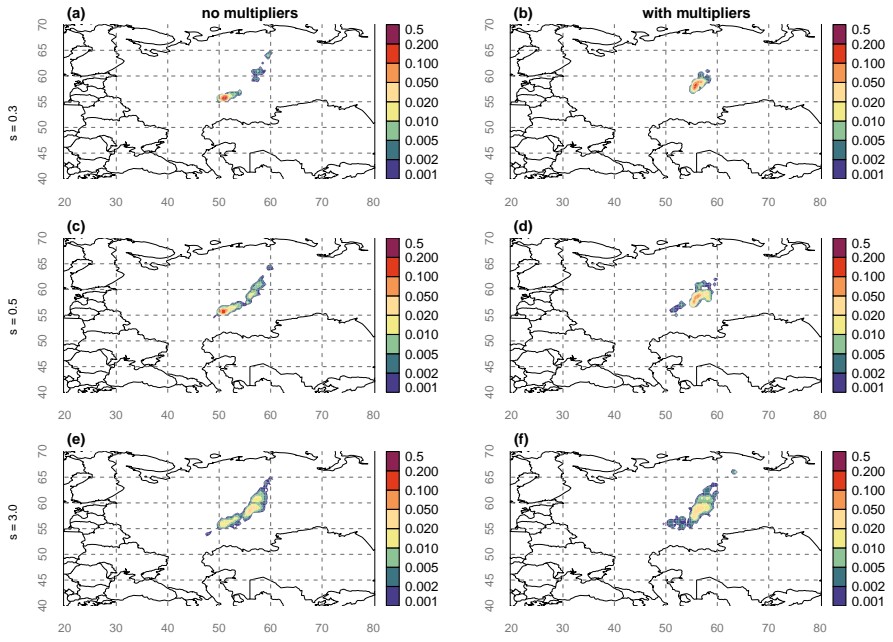

**Figure 3.** Source location probability maps obtained from the Bayesian source reconstruction. The model uncertainty parameters of the inverse gamma distribution were fixed *a priori*, and different values for $s$ were used in the panels: $s = 0.3$ in (a, b), $s = 0.5$ in (c, d) and $s = 3.0$ in (e, f). No multipliers were used in (a, c, e), while multipliers were used in panels (b, d, f) (see Subsection 4.3 for details). The longitude and latitude are shown by gray numbers and dashed lines.

## 4.2 Introduction to multipliers

In this and the next subsection, an alternative model error structure will be discussed involving multipliers or scale factors. Besides being an alternative model error, multipliers could also be used to take into account unknown errors (such as errors due to local atmospheric features not resolved by the model).

240    Yee et al. (2014) used a small number of activity concentration measurements to retrieve the known source parameters of a major medical isotope production facility. The resulting source longitude, latitude and the release term were compared with the true source parameters in order to evaluate the performance of the source reconstruction. The method they used is similar to the method employed here: a Lagrangian stochastic particle model was used in backward mode to generate the source-receptor-sensitivities, and the source parameters were obtained using Bayesian inference and the MT-DREAM$_{(ZS)}$ algorithm. Yee et al.

245    (2014) showed that the main challenge in source reconstruction lies in the correct specification of the model uncertainty (both scale and structure) by using two different measurement models:





$$c_{obs,i} = c_{mod,i} + \epsilon_i \qquad (19)$$

$$c_{obs,i} = m_i \, c_{mod,i} + \epsilon_i' \qquad (20)$$

In Eq. 19, $\epsilon_i$ is the combined model and measurement error, which is assumed to be drawn from a Gaussian distribution with
unknown standard deviation. Again, the unknown standard deviation is replaced by a distribution rather than a single fixed
value. In Eq. 20, $\epsilon_i'$ only represents the measurement error. The model error is now taken into account by so-called scale factors
or multipliers $m_i$. These multipliers are parameters of the Bayesian inference and they are allowed to vary between 0.1 and 10.
The range is chosen so that the multipliers can compensate for model underpredictions or overpredictions of up to a factor 10.

Yee et al. (2014) found that using Eq. 19 resulted in a source reconstruction that did not include the correct source parameters.
The multipliers (Eq. 20) on the other hand resulted in a huge shift in the posterior source location, which significantly improved
the source reconstruction.

### 4.3 Multipliers as unknown model error

Here, multipliers ($m_i$) are introduced to account for unknown model uncertainties that are not yet taken into account by the
likelihood formulation. For instance, local atmospheric features which are not resolved by the model might result in incorrect
source-receptor-sensitivities. Such errors are very hard to quantify, because the computational power to resolve such features
is prohibitively high (at least for continental-scale problems).

The multipliers are additional parameters that need to be estimated during the Bayesian inference. For the sampling of the
parameter, we work with $\log_{10}(m_i)$ and assign a uniform prior between $[-1, 1]$. While the multipliers increase the run time
of the Bayesian inference, the increase is small - especially if one considers the huge increase in the number of unknown
parameters (one additional parameter per observation).

We apply the multipliers using three different values for the model uncertainty parameter $s_i$ (0.3, 0.5 and 3), and show the
results in Fig. 3 (b, d, f). Somewhat similar to the results in Yee et al. (2014), the multipliers cause in a shift in the source
location probability (though not as dramatic). Furthermore, the multipliers do not cause a widening in the posterior source
location. It is interesting to contrast the effect of introducing the multipliers versus increasing the value of the parameter $s_i$:
while the latter result in a shift and a widening of the posterior distribution, the former only result in a shift. While the results
with and without multipliers are significantly different for small values of $s_i$ (Fig. 3, a and b), there is substantial agreement
in the results when using $s_i = 3.0$ (Fig. 3, e and f). This suggests that the effect of introducing multipliers and the effect of
increasing the model uncertainty converges when specifying large model uncertainty.

The results in this section lead to the question: which of the source location maps shown in Fig. 3 is correct. Or more
fundamentally: what is the true structure and scale of the model uncertainty. This will be assessed in the next section using the
ensemble of source-receptor-sensitivities described in Subsection 2.2.



# 5 Fitting uncertainty parameters

## 5.1 The SRS ensemble distribution

In this subsection, it is assessed whether the atmospheric transport model error structure can be obtained from the ensemble of source–receptor-sensitivities (SRS). Note that the ensemble is set up to deal with errors arising from the meteorological input data only. While this type of error arguably adds the largest contribution to the total model error, other sources of model error are not included.

As our ensemble contains 51 members (one unperturbed member and 50 perturbed members), there are 51 SRS values available for each spatiotemporal grid box and each observation. In order to obtain the error structure, the data of all spatial grid boxes is aggregated. This does not necessarily destroy the spatial error correlations in the numerical weather prediction data, since the SRS are the result of an integrated trajectory through the atmosphere associated to a specific observation. The following procedure is applied in order to find the error structure:

1. For each SRS file (associated with a certain observation), and for each spatiotemporal grid box, the ensemble median SRS is calculated; each of the 51 SRS values is scaled by its ensemble median.

2. A Lagrangian particle model can only track a finite number of particles due to computational constraints, and this causes stochastic uncertainty when there are very few particles passing through a geotemporal grid box. However, the SRS variations between ensemble members should represent meteorological uncertainty and should not be impacted by stochastic uncertainty. Therefore, a threshold[1] of $\exp(-20)$ [s] is applied on the median: if the median is smaller than the threshold, all its 51 SRS are omitted from the analysis.

3. The natural logarithm is applied to all SRS since these span many orders of magnitude. If any ensemble member has an SRS equal to 0 for a specific grid box, all its 51 SRS are omitted from the analysis.

As an example, the probability density function of ensemble SRS members around its ensemble median is shown in Fig. 4 for an arbitrary observation and an arbitrary time. It can be seen that most SRS values fall within a factor $\exp(3) \approx 20$ of the ensemble median. Of particular interest is how this density can be approximated by a statistical distribution. In Fig. 4, two distributions are fitted to the density: a Gaussian distribution, and a fit using Eq. 5. Interestingly, while the probability density function can roughly be represented by a Gaussian distribution (effectively being a lognormal distribution with respect to the SRS, since the natural logarithm was applied), its tails are heavier than a Gaussian distribution. On the other hand, a much better fit is obtained using Eq. 5. This distribution describes how the (unknown) true activity concentration is distributed on average around the modelled activity concentration. Similarly, the ensemble gives a distribution of the true SRS around the model predicted SRS. Note that the activity concentrations are simply linear combinations of SRS. This result shows that our choice for the model error structure is supported by the ensemble. This initial result will be elaborated in the next subsection.

---

[1]Since the SRS files are output every three hours, the maximum value for the SRS is 10800 s.



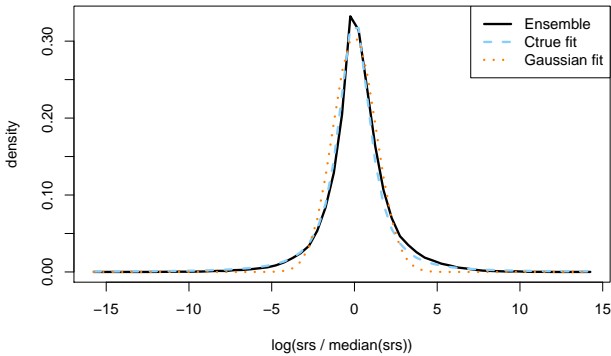

**Figure 4.** Probability density function showing how the atmospheric transport model ensemble members are distribution around the ensemble median (black solid line). Also shown are two fits, one using Eq. 5 (blue dashed line) and one using a Gaussian distribution (orange dotted line). Note that the SRS are scaled by its ensemble median, and that the natural logarithm is applied.

## 5.2 Fitting model uncertainty

In this subsection, the SRS ensemble is used to determine the parameters $s_i$, $\bar{\alpha}_i$ and $\bar{\beta}_i$ in Eq. 5, which describes how the true activity concentration $c_{true}$ is believed to be distributed given the modelled activity concentration $c_{mod,i}$. The fitting can

be performed in several ways, and four different cases are considered. A goodness-of-fit is calculated for each case. As a reference, the fitting procedure is repeated using a Gaussian distribution instead of Eq. 5. The mean of the Gaussian is fixed to be 0 (corresponding to the median of the SRS ensemble), so that only the standard deviation is allowed to vary. Since our Gaussian fitting thus has only one degree of freedom (and given the suggestion of long tails in Fig. 4), it should not be surprising that the Gaussian fit will be outperformed by the fit using Eq. 5.

The following cases are considered to obtain the uncertainty parameters:

- case 1: the parameters are fixed by *a priori* chosen values; for the fit using Eq. 5, we choose $s_i = 0.5$, $\bar{\alpha}_i = 1/\pi$, $\bar{\beta}_i = 1$ for all observations and times; for the Gaussian distribution, we choose $\sigma = 0.5$

- case 2: the parameters are fitted once for all times and all observations (data is aggregated for all times and all observations)

- case 3: the parameters are fitted for each observation (data is aggregated for all times)

- case 4: the parameters are fitted for each observation and each time





The following goodness-of-fit is chosen to quantify how well the fitted probability density function $p_{fit}$ resembles the ensemble probability density function $p_{ens}$; it involves the integration of the absolute difference of both densities and is simply the fraction of overlap in density:


$$\text{overlap in density} = 1 - \frac{1}{2} \int_{-\infty}^{\infty} |p_{ens}(x) - p_{fit}(x)| \, \mathrm{d}x \qquad (21)$$

If the overlap in density equals 1, $p_{fit}$ equals $p_{ens}$. On the other hand, if it equals 0, there is no overlap between $p_{fit}$ and $p_{ens}$.

The uncertainty parameters are obtained using the procedures outlined in case 1 to 4. Then, for each case, the overlap in density is calculated by comparing $p_{fit}$ with $p_{ens}$ for each observation and each time separately, resulting in 288 data points

since there are 24 different times and 12 observations. The set of overlap in densities is used to construct box plots in Fig. 5. The overlap in density is poor when using the *a priori* values for the uncertainty parameters (labels "InvG" and "Gaus", case 1 in Fig 5). While the outcome would have been entirely different if better *a priori* values would have been chosen, its significance should not be underestimated: (i) although seemingly realistic *a priori* uncertainty parameters have been chosen, the agreement with the ensemble density is poor; (ii) for this case and this particular choice of uncertainty parameters, a Gaussian distribution

resulted in a better agreement; (iii) in the absence of an ensemble, using *a priori* chosen uncertainty parameters might be the only available option. When fitting the uncertainty parameters once for all observations and all times, the overlap in density is significantly improved (labels "InvG" and "Gaus", case 2 in Fig 5). The fit using Eq. 5 performs significantly better than the Gaussian fit. Additional yet smaller improvements are obtained when fitting for each observation separately (labels "InvG" and "Gaus", case 3 in Fig 5) and when fitting for each observation and time separately (labels "InvG" and "Gaus", case 4 in Fig 5).

**5.3 Error dependency on the time and the observation**

In this subsection, it is assessed how the fitted uncertainty parameters vary among different observations and different times. The interplay of the three uncertainty parameters $s$, $\bar{\alpha}$ and $\bar{\beta}$ of the inverse gamma function make it less feasible to directly estimate any effect. Therefore, only the fitted standard deviations of the Gaussian distribution are considered here.

The fitted standard deviation for each observation (averaged over all times) is shown in Fig. 6 (left). The standard deviation

varies significantly between the different observations - up to a factor of 2 between the third and the eleventh observation. This can alter the posterior since the uncertainty scales how deviations between the simulated and observed activity concentration are penalized. While it appears difficult to assign meaningful observation-specific *a priori* uncertainty parameters, the ensemble can readily provide such information.

The fitted standard deviation for each time (averaged over all observations) is shown in Fig. 6 (right). It can be seen that the

model uncertainty grows when going backwards in time. The growth is about 30 % during a three day period (note that such growth does not have to be constant over time, and an in-depth assessment is out of the scope of this paper). Also interesting to note is that there is an oscillatory behaviour with a period of eight time steps, corresponding to the diurnal cycle (since





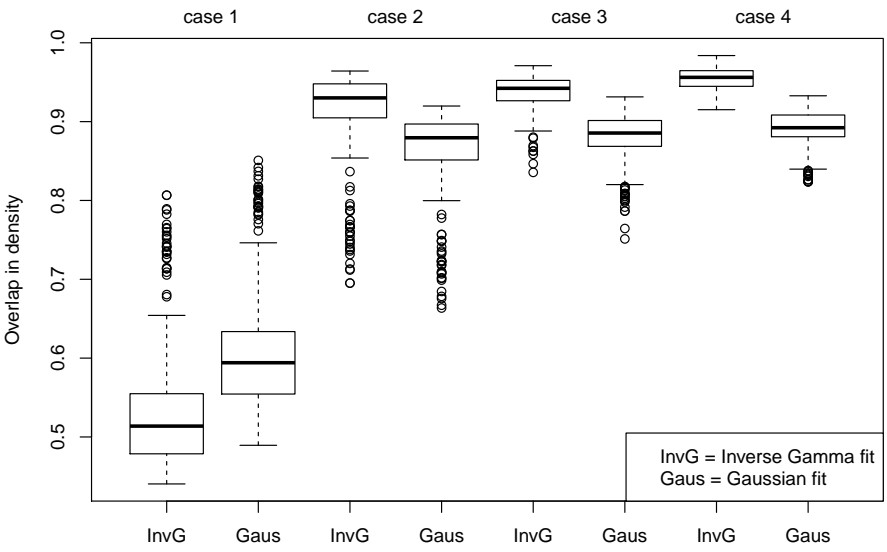

**Figure 5.** Box plots of the overlap between the ensemble densities and the fitted densities using uncertainty parameters obtained in four ways (case 1 to 4, see Subsection 5.2 for details) and using the distribution in Eq. 5 ("InvG") and the Gaussian distribution ("Gaus") for fitting.

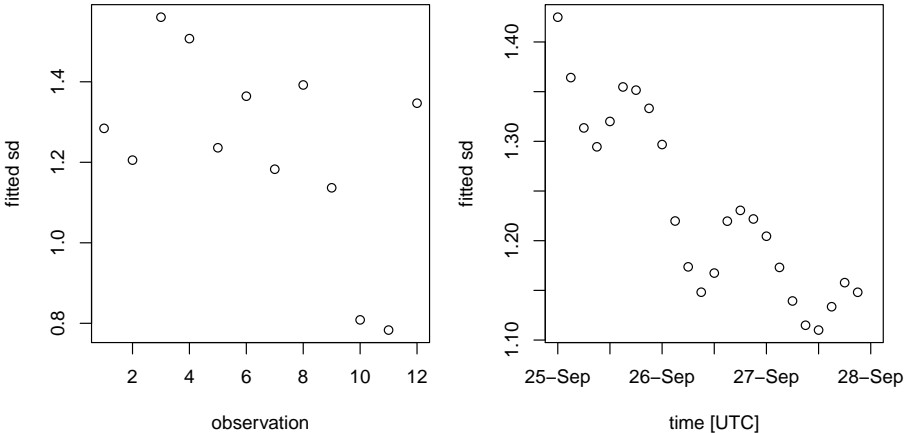

**Figure 6.** The standard deviation obtained after fitting a Gaussian distribution to the SRS ensemble. Left: standard deviation fitted for each of the twelve observations (all times are aggregated); the indices of the observations on the x-axis are arbitrary. Right: standard deviation fitted for each of the twenty-four times (all observations are aggregated); the time step is 3 h.

SRS fields were produced every three hours). The oscillations are likely associated with boundary layer processes, which often follow the diurnal cycle.





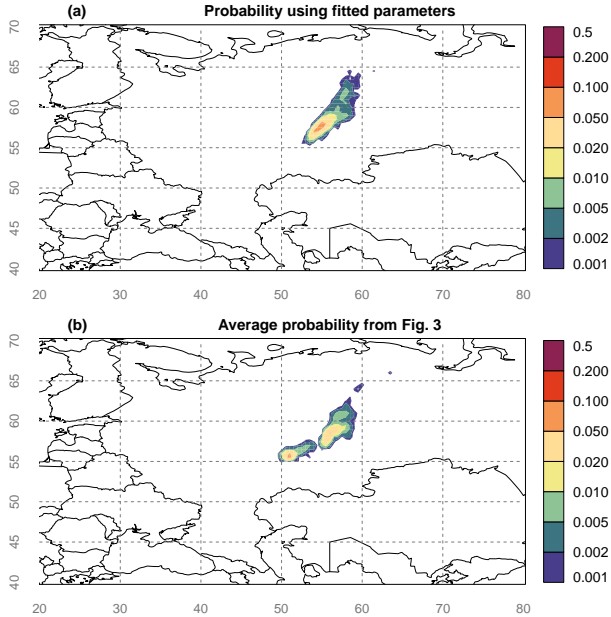

**Figure 7.** Source location probability map obtained from the Bayesian source reconstruction. Top (a): the parameters of the inverse gamma distribution were obtained using the ensemble SRS for each observation separately; bottom (b): the average source location probability is taken from all six panels shown in Fig. 3. The longitude and latitude are shown by gray numbers and dashed lines.

### 5.4 Resulting source location probability map

In this subsection, the Bayesian source reconstruction is applied using fitted observation-specific parameters $s_i$, $\bar{\alpha}_i$ and $\bar{\beta}_i$ (this corresponds to case 3, "invG" in Fig. 5). Furthermore, the inference is run using the SRS ensemble median for each geotemporal grid box - before, the unperturbed ensemble member was used. The resulting probability map is shown in Fig. 7 (a). While the source location probability map is distinct from (most of) the individual panels shown in Fig. 3, it is not too different from what Fig. 3 as a whole suggests. It best resembles case (f) of Fig. 3, which corresponds to the combination of multipliers with large model uncertainty. Note that simply changing the *a priori* uncertainty parameters might never yield a near-perfect correspondence with Fig. 7 (a), since the fitting resulted in different uncertainty parameters for each observation, which seems not possible to obtain without an ensemble.

For comparison, the average probability of the six panels in Fig. 3 is shown in Fig. 7 (b). The latter is perhaps the best approach one can take in the absence of an ensemble. While both maps roughly agree at first glance, there are still important differences. Foremost, the most south-west mode out of two modes in Fig. 7 (b) is absent in Fig. 7 (a). The other mode in Fig. 7 (b) is slightly shifted to the west in Fig. 7 (a), and extends further north-east.





## 6   Ensembles as a set of scenarios

In this section, it is assessed whether additional information can be acquired by considering each ensemble member as an
independent scenario, thus performing the Bayesian source reconstruction for each ensemble member separately. No fitting of
uncertainty parameters is applied here, so that these need to be set *a priori*. The experiment is performed twice, once using
$s_i = 0.5$ and once using $s_i = 3.0$. The other two uncertainty parameters remain fixed ($\bar{\alpha} = 1/\pi$ and $\bar{\beta} = 1$).

### 6.1   Source location probability maps

The Bayesian inference is repeated using the SRS of each ensemble member, so that 51 different posteriors are obtained. These
posteriors then need to be aggregated in some way. As before, we focus on the source location probability. While several
aggregation methods are possible, here the grid box-wise mean and maximum probability is taken (normalisation is required
to ensure that the probabilities sum up to 1). Equal weights were assigned to each ensemble member, since our ensemble is
constructed to yield equally likely scenarios or ensemble members. Note that in the case of multi-model ensembles, the latter
might not be true, so that a weighting should be applied based on the skill of each model.

Fig. 8 (a, b) shows the results using the unperturbed member only, and using $s_i = 0.5$ and $s_i = 3.0$ – hence, these are identical
to panels (c) and (e) in Fig. 3. As discussed earlier in Section 4, the source location probability map differs significantly when
changing the parameter $s$. Fig. 8 (c, d) show the results for the grid box-wise ensemble mean using $s_i = 0.5$ and $s_i = 3.0$. The
results are slightly broader and much smoother, not showing significant bimodal behaviour. Two features are noteworthy: first,
the results roughly approximate those in Fig. 7, though important differences are present, as there were between Fig. 7 (a) and
(b). Second, the results are generally insensitive to the choice of the uncertainty parameter $s$, since Fig. 8 (c) and (d) are very
similar. The same is true for Fig. 8 (e, f), which shows the grid box-wise ensemble maximum. As one can expect, the resulting
source location probability is slightly broader than the one obtained by taking the grid box mean probability; however, the
differences are minor.

It seems that overall, a similar picture is obtained when running the Bayesian inference for each ensemble member separately,
compared to the procedure explained in Section 5. This suggests that if we use the ensemble only (i) to fit the uncertainty
parameters and (ii) to calculate the ensemble median SRS for running the inference as was done in order to obtain Fig. 7, no
crucial information from the ensemble is lost with respect to the source location.

### 6.2   Ensemble convergence

We finally perform a brief assessment on whether or not each ensemble member is adding new information to the ensemble
mean source location probability. For each perturbed member $m \in [1, 50]$, the overlap in source location probability is calcu-
lated between that member and the mean of all previous members $[0, m-1]$ (with member 0 the unperturbed member). We
thus start with the first perturbed member (which is compared with simply the unperturbed member) and end with the last
perturbed member (which is compared with the mean of all other ensemble members). The results are shown in Fig. 9 for an
uncertainty parameter $s = 0.5$ and an uncertainty parameter $s = 3.0$. Note that the overlap in density is calculated using Eq. 21



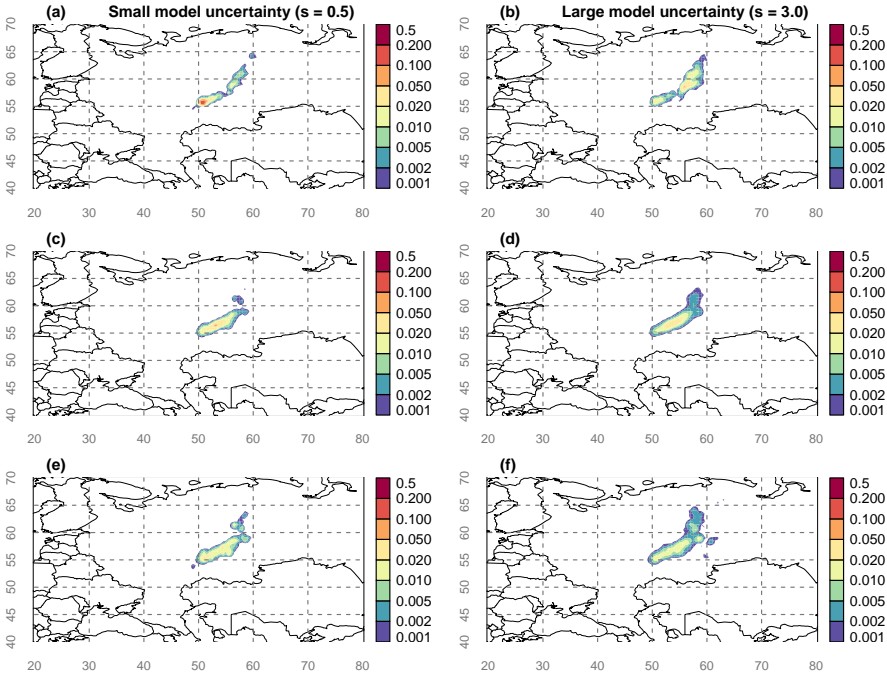

**Figure 8.** Source location probability maps for (a, b): the unperturbed member; (c, d): the grid box-wise ensemble mean; (e, f): the grid box-wise ensemble maximum. Panels (a, c, e) were obtained using $s = 0.5$; panels (b, d, f) were obtained using $s = 3.0$. The longitude and latitude are shown by gray numbers and dashed lines.

and has the same meaning as before, with 0 denoting no overlap (thus being fully informative) and 1 denoting full overlap (providing no new information).

Fig. 9 shows that most ensemble members have an overlap in density less than 0.6. There is significant variance in how much new information is added by each ensemble member. A linear fit suggests that the added information from additional ensemble members is slowly decreasing as expected. Note that the effect is more pronounced for the case with higher model uncertainty
(Fig. 9, right). The reason for this is that the source location probability is more spread out due to the higher model uncertainty, making an overlap more likely. We conclude that all ensemble members are adding new information. This is desirable, as it shows that the ensemble is well-constructed (if members are generally not adding new information, they could be a waste of computational resources).

## 7 Conclusions

Model error has a huge impact on the posterior obtained through Bayesian source reconstruction, a conclusion in agreement with other studies (e.g., Yee et al., 2014). Specifically for the source location, an increase in the scale of the model error resulted in a non-uniform broadening and a shift in the source location probability.



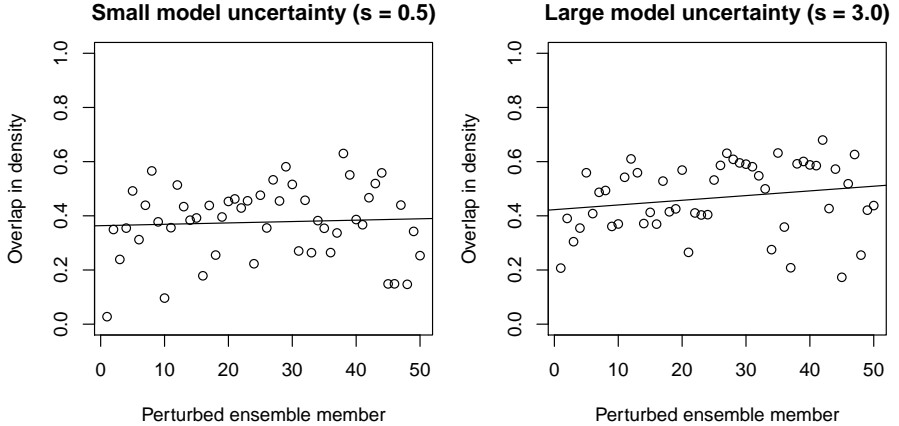

**Figure 9.** Overlap in source location probability as proxy for how much new information each ensemble member is adding to the ensemble mean (see text). Left: results obtained using an uncertainty parameter $s = 0.5$; right: results obtained using an uncertainty parameter $s = 3.0$. A linear fit is also plotted (solid black line).

Both the non-uniformity of the broadening and the shift in the source location probability imply that one cannot simply predict beforehand what the result will be when the Bayesian source reconstruction is repeated with different model error.

In the absence of a way to determine the model error, one could perform multiple Bayesian source reconstructions using different model error formulations as shown in Fig. 3. The results could be aggregated by taking the average (as in Fig 7, b) or by using more elaborate procedures (including a weighted mean).

  Multipliers can be used to represent model error (as in Yee et al., 2014), or to represent the unknown part of model error as was done in Subsection 4.3. The multipliers result in a shift in the source location probability, but not in a broadening.

We found that the ensemble members of source-receptor-sensitivities are distributed around their ensemble median (Fig. 4) in a way that can be well-described by Eq. 5; the latter describes how the true activity concentration $c_{true}$ are assumed to be distributed around the modelled activity concentration $c_{mod}$. Therefore, an ensemble of atmospheric transport and dispersion simulations can be employed to determine the parameters associated with the inverse gamma function in Eq. 16. Of course, the effectiveness of such approach largely depends on whether the ensemble is capable to represent model error.

The ensemble showed that model error varies among different observations (up to a factor 2 in the standard deviation when fitting a Gaussian distribution). Therefore, it is expected that having available model error information which is observation-specific can improve the quality of the Bayesian source reconstruction. The model error is also shown to increase when going further backward in time (for this specific case, there was an increase of 30 % during a three day period in the standard deviation when fitting a Gaussian distribution).

The source location probability using the fitted model error obtained from the ensemble (Fig. 7a) is distinct from the source location probability obtained using fixed uncertainty parameters (individual panels in Fig. 3); however, it is not too different



from what Fig. 3 as a whole suggests, which demonstrates the usefulness of performing multiple Bayesian source reconstructions using different model error formulations as a sensitivity analysis in the absence of an ensemble.

A scenario-based approach (where each ensemble member is used as input for the Bayesian source reconstruction, instead of using the ensemble to fit the uncertainty parameters) gives results which are more robust against the choice of the uncertainty parameters but is more costly compared to directly fitting the uncertainty parameters. No new information is obtained for the source location probability (or stated differently: one does not loose information when using the ensemble only to fit the uncertainty parameters and to calculate the ensemble median for use in the Bayesian inference). The scenario-based approach might be best in case of a small multi-model ensemble, since the fitting of uncertainty parameters might be difficult due to the

difference in skill of each ensemble member.

*Code and data availability.*    The Flexpart model that was used to generate the SRS data is open source and is available for download (Flexpart, 2020). The SRS data from the Flexpart model is available on Zenodo (De Meutter and Delcloo, 2020). The Bayesian inference tool will be made available upon request.

*Author contributions.*    All authors contributed to the conceptualization of the study. PDM conducted the simulations and performed the

analysis. IH and KU supervised the research. All authors contributed to the manuscript. IH took care of the project administration.

*Competing interests.*    The authors declare no competing interests.

*Acknowledgements.*    The authors acknowledge funding from the Defense Research and Development Canada's Canadian Safety and Security Program through project number CSSP-2018-TI-2393.



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
