# Peer review of "On the model uncertainties in Bayesian source reconstruction using an ensemble of weather predictions, the emission inverse modelling system FREARtool v1.0 and the Lagrangian transport and dispersion model Flexpart v9.0.2"

_Geoscientific Model Development, 2020_

## Referee Comment (RC1) · Anonymous Referee #1 · 11 Oct 2020

The proposed manuscript presents important question regarding uncertainties associated with inverse modelling problems for unknown atmospheric releases. It is used ensemble approach which is applayed to the ruthenium 106 case from 2017. The study is interesting and clear to read, but more questions remain in conection with the selected case and its suitability for the cause. Although the goal of the paper is not to find the origin of the ruthenium in 2017, the estimated probability regions of the release far from the actual release site need more discussions. It is difficult to draw and to

follow conclusions based on results inconsistent with state-of-the-art knowledge about the ruthenium case. Perhaps, wider ruthenium dataset or another case with known location and known release profile could be more appropriate for this type of study.

Specific comments:

Although there is dataset with hudreds positive measurements regarding Ru-106 case in 2017 available (Masson et al., 2019), the authors choose data from 5 locations with 12 positive measurements. This is quite suprising and authors should comment this. Moreover, the choice of CTBT stations seems problematic in this case since the main activities have been observed around Mayak and then south-west-wind in Ukraine, Romania etc. (Masson et al., 2019). Hence, my opinion is that the used data can contain rather fractions of information about the release and the results are dominated by the fact that the release period is preselected in the algorithm. This is probably closely related also to the fact that the probable location, Mayak, is not estimated within the probability region in any case (in fact, Dimitrovgrad is much more probable in all cases). This should be discussed in the paper.

p. 6, l. 131: The authors claimed that "the release rate is assumed constant during the release period". This assumption seems to be quite strong since the release rate may vary and, in this particular case of Ru-106 release, did vary during the time as estimated by e.g. (Saunier et al., 2019). Is this assumption necessary and what is the impact of it?

p. 6, l. 141: The authors assume that "the release is assumed to have occurred between 25 September 2017 0000 UTC and 28 September 2017 0000 UTC", however, the release was estimated before e.g. in (Saunier et al., 2019; Western et al., 2020). Could you, please, comment this choice?

p. 6, l. 154: The Currie critical threshold, $L_C$, is used extensively in the paper. Could you please briefly explain basics about this value?

[Figure]

p. 9, l. 203: $\sigma_{obs}$ seems to be fixed in your scenario. Do you have uncertainties associated with measurements? How are these uncertainties related to this $\sigma_{obs}$ value?

Technical corrections:

It is not necessary to have new paragraph after each equation. The abraviation FREAR (in title) is not used and define in the manuscript. p. 8, l. 189: consider to remove "used". p. 10, l. 219: there is no $s_i$ in Eq. (13), please, clarify. Sec. 4.1: you should specify that this is related to the Fig. 3, LEFT.

Literature:

Masson, O., et al.: Airborne concentrations and chemical considerations of radioactive ruthenium from an undeclared major nuclear release in 2017, Proceedings of the National Academy of Sciences, 116:16750–16759, 2019.

O. Saunier, D. Didier, A. Mathieu, O. Masson, and J.D. Le Brazidec. Atmospheric modeling and source reconstruction of radioactive ruthenium from an undeclared major release in 2017. Proceedings of the National Academy of Sciences, 116(50):24991–25000, 2019.

L.M. Western, S.C. Millington, A. Benfield-Dexter, and C.S. Witham. Source estimation of an unexpected release of Ruthenium-106 in 2017 using an inverse modelling approach. Journal of Environmental Radioactivity, 220:106304, 2020.

---

## Referee Comment (RC2) · P. Armand (Referee) · 18 Oct 2020

This paper addresses a question of utmost importance, namely is it possible to determine (better than it is done today) the uncertainties associated with the features of an atmospheric emission? The features of the emission encompass together the release rate, duration and location. This issue is also called "source term estimate" and its applications are numerous especially when a serious nuclear or radiological accident
or the verification of the compliance with the Comprehensive nuclear Test Ban Treaty are at stakes.

In the method developed by the authors, the source term estimate basically uses environmental observations and the FLEXPART atmospheric transport and dispersion model run in backward mode with ECMWF numerical weather predictions as input. Furthermore, this work relies on the Bayesian source reconstruction. Uninformative bounded uniform priors are used to characterize the emission while a general purpose Markov Chain Monte Carlo algorithm, called MT-DREAM(ZS), is used to sample the posterior distribution.

The Bayesian approach has the advantage to providing a probabilistic description of the source term parameters. Thus, the specification of the observation error and of the model error is mandatory, what is the beginning of great difficulties as there is no straightforward way to find out the error of an atmospheric transport and dispersion model. In this regard, the authors use a ensemble of numerical weather predictions in order to create an ensemble of atmospheric transport and dispersion simulations. Then, they propose different methods to infer the model error using these simulations in particular.

The application is carried out for the Ru-106 detections that occurred in September and October 2017 in Eastern Europe and Russia. As pointed out by the authors, their aim is not to identify the release location what has been presumably done in numerous other papers, but to evaluate and compare methods using an ensemble of weather predictions to account for uncertainties in transport and dispersion simulations, and subsequently, source term estimate.

The paper is very interesting, well structured and well written with very few typos. The results obtained by the authors are credible and it is very likely that these results are correct. The comments by the authors are scientifically sound, relevant and mostly convincing.

However, I have some remarks and questions listed hereafter for the authors.

L 21 – What do the the authors call "anomalous radionuclide detections"? That is something I am perfectly aware of, but it is perhaps not the case of all readers.

L 25 – According to the authors, atmospheric transport and dispersion modelling is "one of the methods" to relate detections and the source of emission. I do not see other methods. Which other methods do the authors have in mind?

L 30 – In backward modelling, the source-receptor relationships are calculated from fixed receptors to potential sources (not the opposite as written in the sentence in L 30).

L 32 – The concept of "non-detection" should be explained (or ignored as it is not used in the paper).

L 44 – In this paper, the model error is considered as a whole. Thus, it does not originate only from the numerical weather predictions, but also from the atmospheric transport and dispersion model. The word "mainly" ("because of the underlying weather prediction data") is questionable. The authors should consider rephrasing the sentence.

L 54 – As for me, it is difficult to create and use a relevant ensemble. The reason is not only (and perhaps not mainly) the computational cost of the ensemble, but the way to constitute it with enough variety, limited redundancy, etc. This complex task should be mentioned in the paper.

L 57 – Ditto. It is complicated and not guaranteed that an ensemble captures "most of the possible outcomes". This should be indicated in the paper.

L 59 – What is a "measurement model"?

L 88 – The description of the detections should be gathered in a table with the collection start and stop times (even if I guess that the authors do not wish to develop this aspect

of the data).

L 96 – The beginning of the sentence is "the above observation times". I do not see any observation times above?

L 101 – It is written that FLEXPART is run in backward mode. I wonder how long the simulations go back in time. Could the authors give information about this?

L 110 – It is not obvious that adding and substracting perturbations from an ensemble mean are a legitimate process. Could the authors comment on this?

L 113 – The authors assert that "the spread between the different members represent the uncertainty". This is undoubtedly a way to account for uncertainty in weather predictions, but are the authors sure that the ensemble perfectly encompasses the uncertainty on the meteorological data? The authors should consider being more cautious and rephrasing this sentence.

L 130 – What are the values of t1 and tm, the first and last time for which source-receptor-sensitivities are available for the source reconstruction?

L 131 – The authors assume that the release rate is constant during the release period. I would like to point out that this is a strong assumption as in principle, the release is not known at all. Could the authors comment on this?

L 138 – The total release is assumed to be between 10**10 and 10**16 Bq. This seems to me somewhat arbitrary as it excludes potential releases respectively further downwind and further upwind. Once more, how to proceed when no preconceived solution is available? Could the authors consider commenting on this point?

L 141 – Ditto. How did the authors choose the time interval of the release (all the more that this time interval is quite short)?

L 148 – This is another strong hypothesis that the observations are independent while there is likely a space and time dependency between them. Could the authors comment on this?

L 160 – Does the index "i" in formula (5) indicate that there are as many applications of this formula (with possibly different values of the s, alpha bar and beta bar parameters) as the number of observations?

L 189 – I wonder if the general-purpose Markov Chain Monte Carlo algorithm MT-DREAM(ZS) is freely available? Who developed this MCMC method?

Figure 2 – I suppose that "MDC" stands for "Minimum Detectable Concentration" and that we have LC # MDC / 2. In the formulae, it seems that only LC is used. Could the authors confirm this point?

L 192 – While popular, MCMC methods have well-known drawbacks like the burn-in period or convergence problems. Could the authors consider commenting on this with respect to the MT-DREAM(ZS) algorithm?

L 197 – I have the feeling that all technical details in the last part of this paragraph (and notably the "snooker step") would need some more explanations as this part of the text is too concise (and a bit obscure).

L 209 – It is written here that "s" is an estimate of "sigma", but "sigma" is not defined, nor introduced before. Should the reader understand that sigma stands for sigma_mod?

L 215 – In formula (17), "sigma_srs" and "srs" are not defined. What do these notations stand for? Moreover, what is the reason for the multiplicative value of 16 (and not another value) in the same formula? Could the authors comment on this?

L 216 – The sentence: "as a consequence, the model uncertainty does not depend on the source parameters" is especially unclear or unprecise. What do the authors call "the model"? Is it the weather prediction or the transport and dispersion simulation or both? As the source parameters are not considered as uncertain, I do not see why and how they should take part in the model uncertainty. Please, consider rephrase this sentence.

L 218 – I wonder how "a part of the plume" can be "subject to more atmospheric transport and dispersion processes". All parts of the plume are subject to atmospheric transport and dispersion processes. Small detections may be obtained at the "edge" of the plume or just far from the source of the release. What does a "small" detection mean? It is just a matter of detection method and device. While I globally agree with the ideas contained in this paragraph, I feel that they should be formulated in a different way.

L 226 – The whole section 4 uses the ECMWF unperturbed weather prediction. This should be mentioned at the beginning of the section.

L 229 – As I understand "s_i" includes the model error and the observation error. I wonder what the respective parts of each kind of errors are. Could the authors comment on this? The authors present the source location probability map for three values of "s_i". Of course, it is difficult to choose this parameter and it is the central question which the paper deals with. Is it possible for the authors to motivate the choice of the three "s_i" values? Finally, it is written that "the same value s_i is used for all observations". I wonder why different values of s_i should be associated to the observations as the observation error is by assumption the same for each observation and the model error should depend intrinsically on the model and not on the observation.

Figure 3 – The figure 3 as the following figures seem to me a bit small.

L 237 – I do not see what is an "unknown error"? There are observation errors, representativeness errors or model errors including among others the atmospheric processes not resolved by the model. What is "unknown" is not the type of error, but the value to be attributed to the error.

L 270 – Increasing the value of the parameter s_i results in a shift and an enlargement of the posterior distribution. I wonder why introducing multiplier only results in a shift of the posterior. I suppose that it acts as another way to adjust the posterior without any increase in the level of model uncertainty. Could the authors comment on this?

[Figure]

L 272 – I presume that forcing the model uncertainty with a high value of the parameter s_i predominates against the influence of the multipliers. Do the authors have the same explanation?

L 281 – As for me, it is not so obvious that the errors arising from the meteorological input data have the "largest contribution" to the total model error. Would the atmospheric transport and dispersion model be a "bad model" (what is probably not the case of FLEXPART), the dispersion model error would not be negligible. The authors should perhaps moderate their assessment in L 281.

L 285 – How the data of all grid boxes is aggregated should be more explained. For me, it is not an obvious process.

L 298 – The probability density function of the SRS members should be presented not only for "an arbitrary observation and an arbitrary time" as in Figure 4, but for other observations and times or all distributions should be considered and their moments computed.

Figure 4 – There is a typo in the caption: "distributed" versus "distribution".

L 321 – I wonder about the generality of the method presented by the authors, especially in case 4 when the parameters are fitted for each observation and time. As a matter of fact, it means that just adding or removing a detection will not only influence the source term estimate, but also the uncertainty on this estimate (and this with the same meteorological fields). Could the authors comment on this?

L 347 – Considering "observation-specific" uncertainty parameters is an ad hoc (and interesting) way to fit the model (and observation) error, but it should not be forgotten that the model error should be an intrinsic feature of the model and not depend on the set of observations which is taken into account. I suggest that the authors argue on this.

L 350 – That the model uncertainty grows when going backwards in time is somewhat

trivial. At least, the contrary would be surprising.

L 353 – It is worth noticing that the oscillations have a circadian period. Is it possible to relate them with the day and night alternation of the boundary layer?

L 365 – It is quite optimistic to assert that both maps in Figure 7 roughly agree. There are many differences. Would the location of the release be the aim of the study, the authors would be certainly quite embarrassed to designate it using one map or the other.

L 390 – I would like to point out that there is an interesting result in L 390. As a matter of fact, using the ensemble only to fit the uncertainty parameters or running all members of the ensemble to figure out the uncertainty seems to be equivalent.

L 410 – As a conclusion, I would suggest to the authors to apply the different approaches and methods presented in their paper to situations in which the source characteristics (especially the location) is known unambiguously (because in the Ru-106 case the source location was not really recognized). In a situation with a clearly identified location of the emission, it would be interesting to see what results (good or less good) are obtained using the inference in different ways, and also what is the most efficient approach.

L 435 – As argued by the authors, it seems that using the members of an ensemble in the source term estimate gives more robust results with regard to the choice of the uncertainty parameter as opposed to not using any ensemble. It seems to me quite logical as the ensemble introduces a kind of uncertainty (which is certainly not all the uncertainty, but a "rigorously built" uncertainty). This uncertainty may predominate against the uncertainty arbitrarily fixed by choosing the uncertainty parameter.

---

## Referee Comment (RC3) · Anonymous Referee #3 · 27 Oct 2020

The authors studied the model error estimations and their impact on the inverse modeling using ensemble simulations of 2017 $Ru^{106}$ detections from several CTBT stations. Several interesting findings are presented. The paper is pretty well written.

General comments: While the paper relies on a set of ensemble simulations to quantify the model uncertainties for the emission inverse modeling study, it is helpful to include ensemble in the title.

This paper emphasizes on the spatial patterns of the reconstructed sources. Since the sources also possess the temporal patterns, it is better to describe briefly what the reconstructed sources appear in time. How do the release start time and end time vary with the different approaches in this paper?

Specifics:

Title: FREARtool in the title is never mentioned in text. In the Code and data availability part, it is stated that the "Bayesian inference tool will be made available upon request". If this tool is not mature enough to be available publicly, it is better not to appear in the title.

Line 5: It is not clear what the authors mean by "credible intervals". Is "interval" used to represent the range of emission rates in magnitude? Please clarify this.

Line 103: It is not accurate to say "model output frequency was three hours". In addition, the output can be instantaneous or time-averaged quantities. This needs to be clarified.

Line 105: The emission grid and the concentration grid can be different. Please specify which "grid box" is referred here.

Lines 105-6: Again, it is not accurate to refer the averaging time period as "the output frequency" here.

Lines 107-110: Please specify the resolutions of the meteorological data inputs for FLEXPART.

Lines 138-9, "Since this spans many orders of magnitude, we take log 10(Q) as source parameter in our implementation and simply impose a uniform prior between 10 and 16": Does that mean the accumulated release Q is assumed as $10^{13}$ Bq?

Lines 197-200: These steps are quite important. Brief descriptions of them are suggested here.

Section 5.2: In this section, the use of "time" (e.g. lines 318, 320, and 321) is confusing. I believe it is used to refer the chosen 3-hr release time intervals. Please clarify.

---

## Short Comment (SC1) · 27 Oct 2020

Dear authors,

in my role as Executive editor of GMD, I would like to bring to your attention our Editorial version 1.2:

https://www.geosci-model-dev.net/12/2215/2019/

This highlights some requirements of papers published in GMD, which is also available on the GMD website in the 'Manuscript Types' section: http://www.geoscientific-model-development.net/submission/manuscript_types.html

In particular, please note that for your paper, the following requirement has not been met in the Discussions paper:

- Code must be published on a persistent public archive with a unique identifier for the exact model version described in the paper or uploaded to the supplement, unless this is impossible for reasons beyond the control of authors. All papers must include a section, at the end of the paper, entitled "Code availability". Here, either instructions for obtaining the code, or the reasons why the code is not available should be clearly stated. It is preferred for the code to be uploaded as a supplement or to be made available at a data repository with an associated DOI (digital object identifier) for the exact model version described in the paper. Alternatively, for established models, there may be an existing means of accessing the code through a particular system. In this case, there must exist a means of permanently accessing the precise model version described in the paper. In some cases, authors may prefer to put models on their own website, or to act as a point of contact for obtaining the code. Given the impermanence of websites and email addresses, this is not encouraged, and authors should consider improving the availability with a more permanent arrangement. Making code available through personal websites or via email contact to the authors is not sufficient. After the paper is accepted the model archive should be updated to include a link to the GMD paper.

Therefore your statement in the Code Availability Section "The Bayesian inference tool willbe made available upon request" is not sufficient for a GMD publication. The code needs to be made publicly available in a permanent archive. This is a precondition for the final publication of the article, unless there are license issues to prevent this. In this
case these license issues need to be stated clearly in the code availablility section.

Best regrads,

Astrid Kerkweg (Executive Editor)

---

## Author Comment (AC1) · 6 Nov 2020

**Reply to Anonymous Referee #1**

Dear Reviewer 1,

We would like to thank you for your review. We believe that your comments and suggestions will help us to improve our manuscript. Please find below a step-by-step reply to your comments and suggestions.

Yours faithfully,
The authors

*The proposed manuscript presents important question regarding uncertainties associated with inverse modelling problems for unknown atmospheric releases. It is used ensemble approach which is applayed to the ruthenium 106 case from 2017. The study is interesting and clear to read, but more questions remain in conection with the selected case and its suitability for the cause. Although the goal of the paper is not to find the origin of the ruthenium in 2017, the estimated probability regions of the release far from the actual release site need more discussions. It is difficult to draw and to follow conclusions based on results inconsistent with state-of-the-art knowledge about the ruthenium case. Perhaps, wider ruthenium dataset or another case with known location and known release profile could be more appropriate for this type of study.*

**Specific comments:**

*Although there is dataset with hudreds positive measurements regarding Ru-106 case in 2017 available (Masson et al., 2019), the authors choose data from 5 locations with12 positive measurements. This is quite suprising and authors should comment this. Moreover, the choice of CTBT stations seems problematic in this case since the main activities have been observed around Mayak and then south-west-wind in Ukraine, Romania etc. (Masson et al., 2019). Hence, my opinion is that the used data can contain rather fractions of information about the release and the results are dominated by the fact that the release period is preselected in the algorithm. This is probably closely related also to the fact that the probable location, Mayak, is not estimated within the probability region in any case (in fact, Dimitrovgrad is much more probable in all cases). This should be discussed in the paper.*

**Reply:**

There are several reasons for using only a small subset of all the available data:

1. Many observations have redundant information so that including these additional observations would not significantly affect the results.

2. If we would have used observations taken nearby the Mayak facility, indeed the source location could be much more confined. However, the meteorological data and the EDA ensemble that were used in this study are not appropriate for use on local scales, as the ensemble does not capture well local scale uncertainties.

3. In the CTBTO verification context, we typically have available sparse observations around the world at receptor locations from the International Monitoring System that are about O(1,000 km)

apart from each other. Hence, we are dealing with long-range inverse atmospheric transport problems. As this work was carried out in the context of the CTBT, we have chosen to use (a subset of) observations from the IMS.

4. Furthermore, our results are in agreement with other inverse long-range atmospheric transport modelling studies as it shows a broad region in the Southern Urals as possible source location (see for instance Fig 1 of Saunier et al 2019).

5. Another point to take into account is the computational effort associated to ensemble atmospheric transport modelling. If we would use 1,000 observations, this would require 51,000 Flexpart simulations if we assume that the source location is unknown.

We thank you for your comment since indeed we did not discuss these reasons in our original manuscript. We will add the above discussion in the revised manuscript.

> *p. 6, l. 131: The authors claimed that "the release rate is assumed constant during the release period". This assumption seems to be quite strong since the release rate may vary and, in this particular case of Ru-106 release, did vary during the time as estimated by e.g. (Saunier et al., 2019). Is this assumption necessary and what is the impact of it?*

**Reply:**

It is important to distinguish between different geotemporal scales. While time-varying emissions can have a huge impact nearby the source, these effects are less significant further away from the source due to the atmospheric transport and dispersion processes (and the atmospheric transport model, which filters such information out). Hence, we expect a constant release within release parameters t_start and t_stop to be appropriate to describe the Ru-106 source. In the version of the FREARtool that is described here, the source is parameterised as a fixed release during a certain period. A more recent version of the FREARtool can also deal with time-varying emissions.

> *p. 6, l. 141: The authors assume that "the release is assumed to have occurred between 25 September 2017 0000 UTC and 28 September 2017 0000 UTC", however,the release was estimated before e.g. in (Saunier et al., 2019; Western et al., 2020).Could you, please, comment this choice?*

**Reply:**

According to Figure 2 of Saunier et al 2019, the release that occurred between 25 and 28 September 2017 is about three orders of magnitude larger than the release outside that time period. Since our source parameterisation deals with a fixed release rate between a certain period, and we do not consider local observations which might be affected by smaller releases that potentially took place before 25 September, we believe it is appropriate to neglect smaller emissions that potentially occurred earlier (or later).

> *p. 6, l. 154: The Currie critical threshold,LC, is used extensively in the paper. Could you please briefly explain basics about this value?*

**Reply:**
We will add to the revised manuscript:

"Currie (1968) defined a critical level L_C above which a net signal (which is the detected signal from which the effects of background radiation are subtracted) should be in order to declare the net signal to be a detection."

Note that, without statistical fluctuations, one could simply use L_C = 0, since any positive net signal would be a detection. However, as this is not the case, we set L_C > 0 in order to reduce the number of false positives.

> *p. 9, l. 203: σobs seems to be fixed in your scenario. Do you have uncertainties associated with measurements? How are these uncertainties related to this σobs value?*

**Reply:**

$\sigma_{obs}$ is specific for each observation and is the measurement uncertainty reported by the International Monitoring System. However, there was an inaccuracy in L 203 of the original manuscript, where we wrote:

> "The detected activity concentration $c_{det}$ is assumed to be Gaussian distributed around the true activity concentration $c_{true}$, with a standard deviation $\sigma_{obs} = L_C / k_\alpha$ as in Eq. 14."

This is only valid in case of a true non-detection or a false alarm. We have rewritten that sentence as follows:

> "The detected activity concentration $c_{det}$ is assumed to be Gaussian distributed around the true activity concentration $c_{true}$, with a standard deviation $\sigma_{obs}$ equal to the reported observation error."

The latter makes it also more clear that $\sigma_{obs}$ is not a fixed value.

**Technical corrections:**

> *It is not necessary to have new paragraph after each equation.*

**Reply:**

Thank you for pointing this out.

> *The abraviation FREAR(in title) is not used and define in the manuscript.*

**Reply:**

Thank you for pointing this out. We will add to the revised manuscript:
"FREAR stands for Forensic Radionuclide Event Analysis and Reconstruction"

> *p. 8, l. 189: consider to remove"used".*

**Reply:**

Thank you for your suggestion.

*p. 10, l. 219: there is no si in Eq. (13), please, clarify.*

**Reply:**

Thank you for pointing this out. Eq. (13) is simply Eq. (5) but with c_true replaced by c_det,i. We will also add a reference to Eq. (5) in the manuscript for clarity.

*Sec. 4.1: you should specify that this is related to the Fig. 3, LEFT*

**Reply:**

Indeed, thank you for this remark. We will specify this in the revised manuscript.

---

## Author Comment (AC2) · 6 Nov 2020

**Reply to Referee #2**

Dear Reviewer 2,

We would like to thank you for your in-depth review and interesting thoughts. We believe that your comments and suggestions will help us to improve our manuscript. Please find below a step-by-step reply to your comments and suggestions.

Yours faithfully,
The authors

*L 21 – What do the the authors call "anomalous radionuclide detections"? That is something I am perfectly aware of, but it is perhaps not the case of all readers.*

**Reply:**

Thank you for this suggestion. We will add the following definition to the manuscript:

"Anomalous radionuclide detections are detections of anthropogenic radionuclides originating from upwind nuclear facilities, where the detected concentration of (a) specific radionuclide(s) and/or the combination of several detected radionuclides are anomalous with respect to the station's detection history and/or with respect to what can be expected from these upwind nuclear facilities operating under normal conditions."

*L 25 – According to the authors, atmospheric transport and dispersion modelling is "one of the methods" to relate detections and the source of emission. I do not see other methods. Which other methods do the authors have in mind?*

**Reply:**

In theory, ratios of specific radionuclides (if these are all detected in a certain sample, and assuming no contamination from other sources) could help to discriminate between different sources, without using an atmospheric transport model.

*L 30 – In backward modelling, the source-receptor relationships are calculated from fixed receptors to potential sources (not the opposite as written in the sentence in L 30).*

**Reply:**

We will correct this ambiguity in the revised manuscript.

*L 32 – The concept of "non-detection" should be explained (or ignored as it is not used in the paper).*

**Reply:**

We will add to the revised manuscript (in green):

"Statistical methods can then be employed to combine the information from all these detections (and possibly non-detections - observations where the activity concentration is below a minimum detectable concentration) in a meaningful way in order to infer relevant information on the source."

*L 44 – In this paper, the model error is considered as a whole. Thus, it does not originate only from the numerical weather predictions, but also from the atmospheric transport and dispersion model. The word "mainly" ("because of the underlying weather prediction data") is questionable. The authors should consider rephrasing the sentence.*

**Reply:**

In our experience, the NWP data results in the largest uncertainty in atmospheric transport modelling using the Flexpart model. In Flexpart, the NWP data determines the transport (by the wind) and dispersion (through parameterisation using atmospheric stability) of particles. Source uncertainties are not applicable here, since we work backward in time. In our experience, Flexpart is fairly robust against perturbations of the Flexpart model parameters.

There is also literature that supports our claim in L 44. We will add these references in the revised manuscript:

Engström, A., & Magnusson, L. (2009). Estimating trajectory uncertainties due to flow dependent errors in the atmospheric analysis. *Atmospheric Chemistry & Physics, 9*(22).

Harris, J. M., Draxler, R. R., & Oltmans, S. J. (2005). Trajectory model sensitivity to differences in input data and vertical transport method. *Journal of Geophysical Research: Atmospheres, 110*(D14).

Hegarty, J., Draxler, R. R., Stein, A. F., Brioude, J., Mountain, M., Eluszkiewicz, J., ... & Andrews, A. (2013). Evaluation of Lagrangian particle dispersion models with measurements from controlled tracer releases. *Journal of Applied Meteorology and Climatology, 52*(12), 2623-2637.

However, we welcome findings or literature from the Reviewer that would contradict or complement the above and remain open to adapt that part of the manuscript accordingly.

*L 54 – As for me, it is difficult to create and use a relevant ensemble. The reason is not only (and perhaps not mainly) the computational cost of the ensemble, but the way to constitute it with enough variety, limited redundancy, etc. This complex task should be mentioned in the paper.*

**Reply:**

We agree with that and propose to add the following to the revised manuscript:

"Creating an ensemble with a meaningful spread between its different members (that is, spread which represents the model uncertainty) is a very complex task which requires expert knowledge of all data, processes and their associated uncertainties at each level of the modeling process."

*L 57 – Ditto. It is complicated and not guaranteed that an ensemble captures "most of the possible outcomes". This should be indicated in the paper.*

**Reply:**

We propose the following rewording in the revised manuscript:

"Therefore, ensembles used operationally at major weather institutes around the world are designed in a way that, even with a limited number of members (between 14 and 50, Leutbecher, 2019), the ensemble  tries to capture all (and not more) of the possible outcomes."

*L 59 – What is a "measurement model"?*

**Reply:**

We will refer to Eq. 19 and Eq. 20 in the revised manuscript, and will write that "a measurement model relates the model variable with the observation."

*L 88 – The description of the detections should be gathered in a table with the collection start and stop times (even if I guess that the authors do not wish to develop this aspect of the data).*

**Reply:**

You are correct and this is a helpful suggestion. The text from the paragraph has been reworked to present the data in tabular format.

*L 96 – The beginning of the sentence is "the above observation times". I do not see any observation times above?*

**Reply:**

In that paragraph, we mentioned when the observations were made (L 89 – 92 in the original manuscript). However, we will clarify this in the revised manuscript.

*L 101 – It is written that FLEXPART is run in backward mode. I wonder how long the simulations go back in time. Could the authors give information about this?*

**Reply:**

We will add to the revised manuscript:
"All simulations ended on 20 September 2017."

*L 110 – It is not obvious that adding and substracting perturbations from an ensemble mean are a legitimate process. Could the authors comment on this?*

**Reply:**

This was motivated by the idea that the unperturbed member could perform slightly better than the perturbed members, so that better results could be obtained by centering the perturbations around the unperturbed member rather than around the ensemble mean.

*L 113 – The authors assert that "the spread between the different members represent*

*the uncertainty". This is undoubtedly a way to account for uncertainty in weather pre-dictions, but are the authors sure that the ensemble perfectly encompasses the uncer-tainty on the meteorological data? The authors should consider being more cautious and rephrasing this sentence.*

**Reply:**

With that, we rather meant the general principle of an ensemble: viz. the spread between the members represents the uncertainty. Of course, a bad ensemble will result in a bad uncertainty estimate. We propose to rewrite it as follows:

"The perturbations are created in such a way that each ensemble member represents a possible scenario for the true (unknown) atmospheric state, and the spread between the different members  is simply the model uncertainty as estimated by the ensemble."

*L 130 – What are the values of t1 and tm, the first and last time for which source-receptor-sensitivities are available for the source reconstruction?*

**Reply:**

This is discussed in Subsection 3.2 "prior distribution": $t\_1$ is 25 September 2017 0000 UTC and $t\_m$ is 28 September 2017 0000 UTC. (Flexpart output files were available for other times too.)

*L 131 – The authors assume that the release rate is constant during the release period. I would like to point out that this is a strong assumption as in principle, the release is not known at all. Could the authors comment on this?*

**Reply:**

We repeat our answer to Reviewer 1, who made a similar comment:
"It is important to distinguish between different geotemporal scales. While time-varying emissions can have a huge impact nearby the source, these effects are less significant further away from the source due to the atmospheric transport and dispersion processes (and the atmospheric transport model, which filters such information out). Hence, we expect a constant release within release parameters $t\_{start}$ and $t\_{stop}$ to be appropriate to describe the Ru-106 source."

See also:

De Meutter, P., Camps, J., Delcloo, A., Deconninck, B., & Termonia, P. (2018). Time resolution requirements for civilian radioxenon emission data for the CTBT verification regime. *Journal of environmental radioactivity*, *182*, 117-127.

*L 138 – The total release is assumed to be between 10\*\*10 and 10\*\*16 Bq. This seems to me somewhat arbitrary as it excludes potential releases respectively further downwind and further upwind. Once more, how to proceed when no preconceived solution is available? Could the authors consider commenting on this point?*

**Reply:**

From the available number of measurements, and the scale at which detections were made, these bounds are not unrealistic. Smaller sources would not have been seen over such a broad geographic area, while larger sources would have been seen at more monitoring locations. The selected bounds represent a conservative, but realistic bound for the source. Furthermore, we have already applied inverse modelling using a cost function approach for this case, which allowed us to make our prior distributions sharper than what can be done without knowledge on this case; please see:

De Meutter, P., Camps, J., Delcloo, A., and Termonia, P.: Source Localization of Ruthenium-106 Detections in Autumn 2017 Using Inverse Modelling, in: Mensink C., Gong W., Hakami A. (eds) Air Pollution Modeling and its Application XXVI. ITM 2018. Springer Proceedings in Complexity., Springer, Cham, https://doi.org/10.1007/978-3-030-22055-6_15, 2020.

*L 141 – Ditto. How did the authors choose the time interval of the release (all the more that this time interval is quite short)?*

**Reply:**

(Please also see our reply to your previous comment.) From earlier studies, we knew that the bulk release of Ru-106 likely took place between that period. Since a detailed analysis of the Ru-106 case was not our intention, we have chosen to focus on this time period. An additional benefit of reducing the allowed time interval of the release (when fixing the spatial domain) is that it reduces the memory requirements, which is beneficial when running the case on a personal computer. (Note, however, that the tool can also be run on a server or cluster where more memory is available.)

*L 148 – This is another strong hypothesis that the observations are independent while there is likely a space and time dependency between them. Could the authors comment on this?*

**Reply:**

We acknowledged in the manuscript that this is a simplification. Given the large distance (~ 1000 km) between different IMS stations, we believe this approximation is not too incorrect. Furthermore, the authors are not aware of similar studies that take into account geotemporal dependencies between observations, and we would be grateful if the Reviewer could provide some references.

*L 160 – Does the index "i" in formula (5) indicate that there are as many applications of this formula (with possibly different values of the s, alpha bar and beta bar parameters) as the number of observations?*

**Reply:**

Eq. 5 is indeed for a single observation. The values for s, alpha bar and beta bar can be made observation-specific (which is also done further in the paper).

*L 189 – I wonder if the general-purpose Markov Chain Monte Carlo algorithm MT-DREAM(ZS) is freely available? Who developed this MCMC method?*

**Reply:**

It was developed by Laloy and Vrugt and described in their paper Laloy and Vrugt (2012). Some implementations of DREAM can be found in open source packages on the internet.

*Figure 2 – I suppose that "MDC" stands for "Minimum Detectable Concentration" and that we have LC # MDC / 2. In the formulae, it seems that only LC is used. Could the authors confirm this point?*

**Reply:**

In the formulae, $L_C$ is used. With the observations, typically the MDC is reported and not $L_C$. For the observations in the IMS network of CTBTO, we can assume that $L_C = MDC/2$.

*L 192 – While popular, MCMC methods have well-known drawbacks like the burn-in period or convergence problems. Could the authors consider commenting on this with respect to the MT-DREAM(ZS) algorithm?*

**Reply:**

This depends on the case, but from the authors' experience over the past year, we typically run the tool using ~ 10,000 iterations and convergence occurs after ~ 2,500 iterations (where we discard these first 2,500 iterations). In our previous study however, (De Meutter and Hoffman, 2020) where we studied the Se-75 release, we used 150,000 iterations.
The required number of iterations is also affected by the choice of the uncertainty "s": lower uncertainties require more iterations before convergence takes place.

*L 197 – I have the feeling that all technical details in the last part of this paragraph (and notably the "snooker step") would need some more explanations as this part of the text is too concise (and a bit obscure).*

**Reply:**

Regarding the snooker step, we were informed by one of the developers of MT-DREAM(ZS) that the snooker step is theoretically not compatible with the multiple-try part of the algorithm, so that we no longer use the snooker step. The difference in the posterior after using and not using the snooker step is not noticeable in our simulations.

To prove the latter, please find the results below for two simulations for the Ru-106 case, with and without the snooker step:

1/ simulation with the snooker step for the unperturbed member and s_i = 0.5

```
 Running MT-DREAMzs, iteration 7800 of 50001 . Current logp -37.44259 -41.24711 -39.54531
Converged after 7800 iterations
 Running MT-DREAMzs, iteration 50001 of 50001 . Current logp -36.48064 -44.17845 -41.44014
 MT-DREAMzs terminated after 1206.558 seconds
Acceptance rate for chain 1 is 22.24%
Acceptance rate for chain 2 is 22.61%
Acceptance rate for chain 3 is 22.93%
          lon     lat  log10_Q            rstart               rstop
0.025 50.11799 55.50922 14.96976 2017-09-25 00:22:32 2017-09-26 23:34:40
0.5   51.09007 55.91466 15.27527 2017-09-25 07:59:14 2017-09-27 18:17:46
0.975 57.88037 60.75305 15.64360 2017-09-25 22:55:13 2017-09-27 23:34:05
mean  51.98106 56.39979 15.28396 2017-09-25 08:46:06 2017-09-27 16:41:15
```

2/ simulation without the snooker step for the unperturbed member and s_i = 0.5

```
 Running MT-DREAMzs, iteration 12300 of 50001 . Current logp -44.62895 -44.45257 -41.44722
Converged after 12300 iterations
 Running MT-DREAMzs, iteration 50001 of 50001 . Current logp -43.13082 -39.71556 -37.64025
 MT-DREAMzs terminated after 1271.046 seconds
Acceptance rate for chain 1 is 25.26%
Acceptance rate for chain 2 is 24.75%
Acceptance rate for chain 3 is 23.59%
          lon     lat  log10_Q            rstart                rstop
0.025 50.09579 55.51231 14.97063 2017-09-25 00:30:47 2017-09-26 23:01:53
0.5   51.03933 55.88155 15.26998 2017-09-25 08:00:29 2017-09-27 18:35:54
0.975 57.74679 60.13044 15.65440 2017-09-25 22:38:03 2017-09-27 23:37:18
mean  51.85432 56.30864 15.27988 2017-09-25 08:42:32 2017-09-27 16:59:44
```

The following information will be added to the revised manuscript:

 To enhance efficiency and to obtain more accurate results, randomized subspace sampling is used (Vrugt et al., 2009). This simply means that not necessarily all source parameters are updated at a time, but instead a randomized subset of the source parameters. Furthermore, MT-DREAM (ZS) makes use of multiple try Metropolis sampling (Liu et al., 2000) to enhance the mixing of the chains. This means in practice that, to advance to Markov chain, several proposals are drawn instead of one proposal in traditional Metropolis sampling. Furthermore, the Metropolis acceptance is calculated in a different way (Liu et al., 2000 , Laloy and Vrugt, 2012)."

*L 209 – It is written here that "s" is an estimate of "sigma", but "sigma" is not defined, nor introduced before. Should the reader understand that sigma stands for sigma_mod?*

**Reply:**

Thank you for pointing this out. In L 209, "sigma" should have been "sigma_mod". Note however that in L 222, "sigma" stands for sqrt(sigma_mod^2 + sigma_obs^2). We will add that to the revised manuscript.

*L 215 – In formula (17), "sigma_srs" and "srs" are not defined. What do these notations stand for? Moreover, what is the reason for the multiplicative value of 16 (and not another value) in the same formula? Could the authors comment on this?*

**Reply:**

Thank you for pointing that out. "srs" stands for source-receptor-sensitivities (the model output when Flexpart is run in backward mode), and "sigma_srs" is its (unknown) uncertainty. We will add that to the revised manuscript.

The value of 16 is an empirical number that was found to give a good balance between information obtained from detections versus information from non-detections from an earlier case study described in De Meutter and Hoffman (2020). We will add this information in the revised manuscript.

*L 216 – The sentence: "as a consequence, the model uncertainty does not depend on the source parameters" is especially unclear or unprecise. What do the authors call "the model"? Is it the weather prediction or the transport and dispersion simulation or*

*both? As the source parameters are not considered as uncertain, I do not see why and how they should take part in the model uncertainty. Please, consider rephrase this sentence.*

**Reply:**

We can calculate the modeled activity concentrations c_mod as a linear relation between the source-receptor-sensitivities (srs) and the release amount Q:

c_mod = srs(release period, release location) * Q

One way of calculating the uncertainty on c_mod (sigma_c_mod) would then be to use sigma_srs, which could be obtained from the ensemble:

sigma_c_mod = sigma_srs(release period, release location) * Q

However, in that case, sigma_c_mod will depend on the source parameters (the release period, the release location and Q). This resulted in undesired effects in the very beginning of the development of the FREARtool (such as: the model selecting very high Q so that the uncertainty became very large, thereby allowing values of c_mod that did not agree at all with c_obs), so that it was decided to make the model uncertainty independent of the source parameters.

To avoid confusion, we propose to omit this sentence.

*L 218 – I wonder how "a part of the plume" can be "subject to more atmospheric transport and dispersion processes". All parts of the plume are subject to atmospheric transport and dispersion processes. Small detections may be obtained at the "edge" of the plume or just far from the source of the release. What does a "small" detection mean? It is just a matter of detection method and device. While I globally agree with the ideas contained in this paragraph, I feel that they should be formulated in a different way.*

**Reply:**

We propose the following revision:
L 218: "This is desirable since small detections are caused by a part of the plume of radionuclides that was subject to more atmospheric  dilution..."

*L 226 – The whole section 4 uses the ECMWF unperturbed weather prediction. This should be mentioned at the beginning of the section.*

**Reply:**

Indeed, thank you for pointing this out.

*L 229 – As I understand "s_i" includes the model error and the observation error. I wonder what the respective parts of each kind of errors are. Could the authors comment on this? The authors present the source location probability map for three values of "s_i". Of course, it is difficult to choose this parameter and it is the central question which the paper deals with. Is it possible for the authors to motivate the choice of the three "s_i" values? Finally, it is written that "the same value s_i is used for all observations". I wonder why different values of s_i should be associated to the observations as the*

*observation error is by assumption the same for each observation and the model error should depend intrinsically on the model and not on the observation.*

**Reply:**

The interpretation of the different s_i values is straightforward from Eq. 17: it represents a relative error of 30 %, 50 % and 300 % with respect to max(c_det, 16 * L_C). 50 % was our initial "default value". 300 % seemed a good value to go above that (we also tested other values, such as 100 % and 1 000 %). The choice for the lower value is limited by the observation error (lower values for sigma_total would imply an imaginary model error). Furthermore, some members had troubles with convergence when very small s_i values were chosen (10 %).

The observation error is different for each observation as it depends on the background radiation, the sampled volume of air etc… We believe that the model error should also be observation-specific, please see our reply further below to a comment regarding L 347.

*Figure 3 – The figure 3 as the following figures seem to me a bit small.*

**Reply:**

We will increase the figure size in the revised manuscript.

*L 237 – I do not see what is an "unknown error"? There are observation errors, representativeness errors or model errors including among others the atmospheric processes not resolved by the model. What is "unknown" is not the type of error, but the value to be attributed to the error.*

**Reply:**

In the revised manuscript, we will make the following change:

""

"Besides being an alternative model error, multipliers could also be used to take into account errors that were not fully captured by the model (such as errors due to local atmospheric features not resolved by the model, measurement errors due to sample inhomogeneity, etc.)"

*L 270 – Increasing the value of the parameter s_i results in a shift and an enlargement of the posterior distribution. I wonder why introducing multiplier only results in a shift of the posterior. I suppose that it acts as another way to adjust the posterior without any increase in the level of model uncertainty. Could the authors comment on this?*

**Reply:**

That sounds certainly plausible. The model uncertainty is indeed not affected by the multipliers. The multipliers allow a better match between "m * c_mod" and the observations "c_obs". This better agreement can in theory be obtained with the same source parameters when no multipliers are used (thus, no shift will be seen), or it can be obtained with different source parameters (so that a shift will be seen if the source location is affected).

*L 272 – I presume that forcing the model uncertainty with a high value of the parameter s_i predominates against the influence of the multipliers. Do the authors have the same explanation?*

**Reply:**

If the model uncertainties (determined by s_i) are larger than "|c_mod – c_obs|", then indeed the multipliers will have less impact on the posterior.

*L 281 – As for me, it is not so obvious that the errors arising from the meteorological input data have the "largest contribution" to the total model error. Would the atmospheric transport and dispersion model be a "bad model" (what is probably not the case of FLEXPART), the dispersion model error would not be negligible. The authors should perhaps moderate their assessment in L 281.*

**Reply:**

Please see also our reply to a related comment concerning L 44. We propose the following (minor) moderation but remain open to consider further moderation if the Reviewer could share findings or literature that shows its necessity.

"While this type of error  likely adds the largest contribution to the total model error, other sources of model error are not included."

*L 285 – How the data of all grid boxes is aggregated should be more explained. For me, it is not an obvious process.*

**Reply:**

We will add the following in the revised manuscript (in green):

"In order to obtain the error structure, the data of all spatial grid boxes is aggregated into an uncertainty distribution."

Furthermore, we will add to the list:

> "4. The remaining data points are used to make an uncertainty distribution (as in Figure 4)."

*L 298 – The probability density function of the SRS members should be presented not only for "an arbitrary observation and an arbitrary time" as in Figure 4, but for other observations and times or all distributions should be considered and their moments computed.*

**Reply:**

It is not feasible to plot the distribution for each time and each observation (288 in total) in one figure, but we will add this as supplementary information.

*Figure 4 – There is a typo in the caption: "distributed" versus "distribution".*

**Reply:**

Thank you for noticing this. We will correct this in the revised manuscript.

*L 321 – I wonder about the generality of the method presented by the authors, especially in case 4 when the parameters are fitted for each observation and time. As a matter of fact, it means that just adding or removing a detection will not only influence the source term estimate, but also the uncertainty on this estimate (and this with the same meteorological fields). Could the authors comment on this?*

**Reply:**

It is not only the meteorological fields that determine the uncertainty, but also the trajectories that particles follow along these meteorological fields. As a result, the model uncertainty is observation-specific, and indeed, adding or removing observations can alter the uncertainty on the inferred parameters. See also our next reply.

*L 347 – Considering "observation-specific" uncertainty parameters is an ad hoc (and interesting) way to fit the model (and observation) error, but it should not be forgotten that the model error should be an intrinsic feature of the model and not depend on the set of observations which is taken into account. I suggest that the authors argue on this.*

**Reply:**

We do not agree that the model error should be an intrinsic feature of the model and does not depend on the observation: the model uncertainty depends on the trajectory of the retro-plume (= the plume that goes from the sampling station backward in time). Observations of a plume that are made three weeks after the release should have higher model uncertainty than observations made two days after the release. Also, depending on the weather conditions along the trajectory, the model error can be observation specific (consider transport associated with a frontal system versus transport associated with the calm conditions found in an anticyclone).

To clarify this, we will add (see text in green) to the revised manuscript:
L341: "In this subsection, it is assessed how the fitted uncertainty parameters vary among different observations and different times. The motivation for this is as follows: first, and somewhat trivial, we can expect the model uncertainty to increase as a function of simulation time. Second, uncertainties are expected to be observation-dependent, since observations are made on different times and at different distances from the source; uncertainties on the trajectories between the receptor and the source will also be affected by the atmospheric conditions along the trajectory, which are expected to be observation-specific."

*L 350 – That the model uncertainty grows when going backwards in time is somewhat trivial. At least, the contrary would be surprising.*

**Reply:**

We agree, but it is always good to confirm that our ensemble of atmospheric transport simulations replicates evident features.

*L 353 – It is worth noticing that the oscillations have a circadian period. Is it possible to relate them with the day and night alternation of the boundary layer?*

**Reply:**

We believe that L 351 made that notice:
"Also interesting to note is that there is an oscillatory behaviour with a period of eight time steps, corresponding to the diurnal cycle (since SRS fields were produced every three hours). The oscillations are likely associated with boundary layer processes, which often follow the diurnal cycle."

*L 365 – It is quite optimistic to assert that both maps in Figure 7 roughly agree. There are many differences. Would the location of the release be the aim of the study, the authors would be certainly quite embarrassed to designate it using one map or the other.*

**Reply:**

Indeed, but we assume the output of the inference will be interpreted by an expert, who is aware that models have uncertainties, and that even the uncertainties are uncertain. Also take into account that we zoom in into the area of interest. If we would plot the full domain, the differences will appear smaller.

*L 390 – I would like to point out that there is an interesting result in L 390. As a matter of fact, using the ensemble only to fit the uncertainty parameters or running all members of the ensemble to figure out the uncertainty seems to be equivalent.*

**Reply:**

We believe L 389 in the original manuscript mentions this, but we will try to make it more explicit by adding (in green):

"It seems that overall, a similar picture is obtained when running the Bayesian inference for each ensemble member separately, compared to the procedure explained in Section 5. This suggests that if we use the ensemble only (i) to fit the uncertainty parameters and (ii) to calculate the ensemble median SRS for running the inference as was done in order to obtain Fig. 7, no crucial information from the ensemble is lost with respect to the source location. As a consequence, it is equivalent to running the inference with all members of the ensemble separately to determine the uncertainty."

*L 410 – As a conclusion, I would suggest to the authors to apply the different approaches and methods presented in their paper to situations in which the source characteristics (especially the location) is known unambiguously (because in the Ru-106 case the source location was not really recognized). In a situation with a clearly identified location of the emission, it would be interesting to see what results (good or less good) are obtained using the inference in different ways, and also what is the most efficient approach.*

**Reply:**

Thank you for this suggestion, which is in line with the comments made by Reviewer 1. We will add to the conclusions:
"In a future study, we will apply the different approaches and methods presented in this paper to situations in which the source characteristics are known unambiguously. This will help to better evaluate the different approaches proposed in this paper."

*L 435 – As argued by the authors, it seems that using the members of an ensemble in the source term estimate gives more robust results with regard to the choice of the uncertainty parameter as opposed to not using any ensemble. It seems to me quite logical as the ensemble introduces a kind of uncertainty (which is certainly not all the uncertainty, but a "rigorously built" uncertainty). This uncertainty may predominate against the uncertainty arbitrarily fixed by choosing the uncertainty parameter.*

**Reply:**

We agree with that, and propose to add that in the revised manuscript (in green):

"A scenario-based approach (where each ensemble member is used as input for the Bayesian source reconstruction, instead of using the ensemble to fit the uncertainty parameters) gives results which are more robust against the choice of the uncertainty parameters but is more costly compared to directly fitting the uncertainty parameters. This is because the ensemble introduces model uncertainty that may predominate against the uncertainty prescribed by arbitrarily choosing the uncertainty parameter."

[Figure]

[Figure]

[Figure]

[Figure]

[Figure]

[Figure]

[Figure]

[Figure]

[Figure]

[Figure]

[Figure]

[Figure]

[Figure]

[Figure]

[Figure]

[Figure]

[Figure]

[Figure]

[Figure]

[Figure]

[Figure]

[Figure]

[Figure]

[Figure]

[Figure]

[Figure]

[Figure]

[Figure]

[Figure]

[Figure]

[Figure]

[Figure]

[Figure]

[Figure]

[Figure]

[Figure]

[Figure]

[Figure]

[Figure]

[Figure]

[Figure]

[Figure]

[Figure]

[Figure]

**itime=13, iobs=5**

[Figure]

[Figure]

[Figure]

[Figure]

[Figure]

[Figure]

[Figure]

[Figure]

[Figure]

[Figure]

[Figure]

[Figure]

[Figure]

[Figure]

[Figure]

[Figure]

[Figure]

[Figure]

[Figure]

[Figure]

[Figure]

[Figure]

[Figure]

[Figure]

[Figure]

[Figure]

[Figure]

[Figure]

[Figure]

[Figure]

[Figure]

[Figure]

[Figure]

[Figure]

[Figure]

[Figure]

[Figure]

[Figure]

[Figure]

[Figure]

[Figure]

[Figure]

[Figure]

[Figure]

[Figure]

[Figure]

[Figure]

[Figure]

[Figure]

[Figure]

**itime=17, iobs=7**

[Figure]

[Figure]

[Figure]

[Figure]

[Figure]

[Figure]

[Figure]

[Figure]

[Figure]

[Figure]

[Figure]

[Figure]

[Figure]

[Figure]

[Figure]

[Figure]

[Figure]

[Figure]

[Figure]

[Figure]

[Figure]

[Figure]

[Figure]

[Figure]

[Figure]

[Figure]

[Figure]

[Figure]

[Figure]

[Figure]

[Figure]

[Figure]

[Figure]

[Figure]

[Figure]

[Figure]

[Figure]

[Figure]

[Figure]

[Figure]

[Figure]

[Figure]

[Figure]

[Figure]

[Figure]

[Figure]

[Figure]

[Figure]

[Figure]

[Figure]

[Figure]

[Figure]

[Figure]

[Figure]

[Figure]

[Figure]

[Figure]

[Figure]

[Figure]

[Figure]

[Figure]

[Figure]

[Figure]

[Figure]

[Figure]

[Figure]

[Figure]

[Figure]

[Figure]

[Figure]

[Figure]

[Figure]

[Figure]

[Figure]

[Figure]

[Figure]

[Figure]

**itime=23, iobs=12**

[Figure]

[Figure]

[Figure]

[Figure]

[Figure]

[Figure]

[Figure]

[Figure]

[Figure]

[Figure]

[Figure]

[Figure]

[Figure]

[Figure]

[Figure]

[Figure]

[Figure]

[Figure]

[Figure]

[Figure]

[Figure]

[Figure]

[Figure]

[Figure]

[Figure]

[Figure]

[Figure]

[Figure]

[Figure]

[Figure]

[Figure]

[Figure]

[Figure]

[Figure]

[Figure]

[Figure]

[Figure]

[Figure]

[Figure]

[Figure]

[Figure]

[Figure]

[Figure]

[Figure]

[Figure]

[Figure]

[Figure]

[Figure]

[Figure]

[Figure]

[Figure]

[Figure]

[Figure]

[Figure]

[Figure]

[Figure]

[Figure]

[Figure]

**itime=5, iobs=10**

---

## Author Comment (AC3) · 6 Nov 2020

**Reply to Anonymous Referee #3**

Dear Reviewer 3,

We would like to thank you for your review. We believe that your comments and suggestions will help us to improve our manuscript. Please find below a step-by-step reply to your comments and suggestions.

Yours faithfully,
The authors

**General comments:**

*While the paper relies on a set of ensemble simulations to quantify the model uncertainties for the emission inverse modeling study, it is helpful to include ensemble in the title.*

**Reply:**

Thank you for this suggestion, we propose to revise the title as follows:

"On the model uncertainties in Bayesian source reconstruction using an ensemble of weather predictions, the emission inverse modelling system FREARtool v1.0 and the Lagrangian transport and dispersion model Flexpart v9.0.2"

*This paper emphasizes on the spatial patterns of the reconstructed sources. Since the sources also possess the temporal patterns, it is better to describe briefly what the reconstructed sources appear in time. How do the release start time and end time vary with the different approaches in this paper?*

**Reply:**

It is true that we emphasized on the source location; the reason for doing so is that the release location is of primary interest in the context of CTBT verification. Once a location is found (for instance, based on the location of known nuclear facilities within the posterior source region, or based on a seismic signal associated to a nuclear explosion), a new inference could be performed fixing the release location as was done in De Meutter and Hoffman (2020).

We will add this reasoning to the revised manuscript.

**Specifics:**

*Title: FREARtool in the title is never mentioned in text. In the Code and data availability part, it is stated that the "Bayesian inference tool will be made available upon request". If this tool is not mature enough to be available publicly, it is better not to appear in the title.*

**Reply:**

Thank you for noticing this, we will mention 'FREARtool' in the text. We will make the tool publicly available after a formal announcement during an in-person CTBT-related event, which was unfortunately not possible so far due to the COVID crisis.

*Line 5: It is not clear what the authors mean by "credible intervals". Is "interval" used to represent the range of emission rates in magnitude? Please clarify this.*

**Reply:**

"Credible interval" is a term used in Bayesian inference and represents the range in which an unobservable parameter falls with a particular probability. A credible interval is thus available for each of the five source parameters that are inferred: the source longitude, source latitude, total emission, release start and release end.

*Line 103: It is not accurate to say "model output frequency was three hours". In addition, the output can be instantaneous or time-averaged quantities. This needs to be clarified.*

**Reply:**

Indeed, we will revise this as follows:

"The  time-averaged source-receptor-sensitivities were output every three hours, so that the maximum possible residence time in a geotemporal grid box is 10 800 s."

*Line 105: The emission grid and the concentration grid can be different. Please specify which "grid box" is referred here.*

**Reply:**

Since Flexpart is a Lagrangian particle model, there is no "emission grid": particles are released from point sources, line sources, area sources and / or volume sources, independent of any grid. With "grid box", the Flexpart output grid box is meant. If the simulation goes forward in time, this could be interpreted as a concentration grid. If the simulation goes backward in time, one could call this the emission grid instead.

*Lines 105-6: Again, it is not accurate to refer the averaging time period as "the output frequency" here.*

**Reply:**

We will correct this.

*Lines 107-110: Please specify the resolutions of the meteorological data inputs for FLEXPART.*

**Reply:**

We will add the following in the revised manuscript:

"The EDA system uses a Gaussian grid with 640 latitude lines between pole and equator, but the data was converted to a lon-lat grid having grid spacings of 0.5°."

*Lines 138-9, "Since this spans many orders of magnitude, we take log 10(Q) as source parameter in our implementation and simply impose a uniform prior between 10 and 16": Does that mean the accumulated release Q is assumed as 10 13 Bq?*

**Reply:**

No, the prior distribution from which initial samples are drawn is a uniform distribution between 10 and 16. The accumulated release is thus assumed to be between $10^{10}$ and $10^{16}$ Bq.

*Lines 197-200: These steps are quite important. Brief descriptions of them are suggested here.*

**Reply:**

Regarding the snooker step, we were informed by one of the developers of MT-DREAM(ZS) that the snooker step is theoretically not compatible with the multiple-try part of the algorithm, so that we no longer use the snooker step. The difference in the posterior after using and not using the snooker step is not noticeable in our simulations.

To proof the latter, please find the results below for two simulations for the Ru-106 case, with and without the snooker step:

1/ simulation with the snooker step for the unperturbed member and s_i = 0.5

```
 Running MT-DREAMzs, iteration 7800 of 50001 . Current logp -37.44259 -41.24711 -39.54531
Converged after 7800 iterations
 Running MT-DREAMzs, iteration 50001 of 50001 . Current logp -36.48064 -44.17845 -41.44014
 MT-DREAMzs terminated after 1206.558 seconds
Acceptance rate for chain 1 is 22.24%
Acceptance rate for chain 2 is 22.61%
Acceptance rate for chain 3 is 22.93%
          lon      lat  log10_Q          rstart              rstop
0.025 50.11799 55.50922 14.96976 2017-09-25 00:22:32 2017-09-26 23:34:40
0.5   51.09007 55.91466 15.27527 2017-09-25 07:59:14 2017-09-27 18:17:46
0.975 57.88037 60.75305 15.64360 2017-09-25 22:55:13 2017-09-27 23:34:05
mean  51.98106 56.39979 15.28396 2017-09-25 08:46:06 2017-09-27 16:41:15
```

2/ simulation without the snooker step for the unperturbed member and s_i = 0.5

```
 Running MT-DREAMzs, iteration 12300 of 50001 . Current logp -44.62895 -44.45257 -41.44722
Converged after 12300 iterations
 Running MT-DREAMzs, iteration 50001 of 50001 . Current logp -43.13082 -39.71556 -37.64025
 MT-DREAMzs terminated after 1271.046 seconds
Acceptance rate for chain 1 is 25.26%
Acceptance rate for chain 2 is 24.75%
Acceptance rate for chain 3 is 23.59%
          lon      lat  log10_Q          rstart              rstop
0.025 50.09579 55.51231 14.97063 2017-09-25 00:30:47 2017-09-26 23:01:53
0.5   51.03933 55.88155 15.26998 2017-09-25 08:00:29 2017-09-27 18:35:54
0.975 57.74679 60.13044 15.65440 2017-09-25 22:38:03 2017-09-27 23:37:18
mean  51.85432 56.30864 15.27988 2017-09-25 08:42:32 2017-09-27 16:59:44
```

The following information will be added to the revised manuscript:

" To enhance efficiency and to obtain more accurate results, randomized subspace sampling is used (Vrugt et al., 2009). This simply means that not necessarily all source parameters are updated at a time, but instead a randomized

subset of the source parameters. Furthermore, MT-DREAM (ZS) makes use of multiple try Metropolis sampling (Liu et al., 2000) to enhance the mixing of the chains. This means in practice that, to advance to Markov chain, several proposals are drawn instead of one proposal in traditional Metropolis sampling. Furthermore, the Metropolis acceptance is calculated in a different way (Liu et al., 2000 , Laloy and Vrugt, 2012)."

*Section 5.2: In this section, the use of "time" (e.g. lines 318, 320, and 321) is confusing. I believe it is used to refer the chosen 3-hr release time intervals. Please clarify.*

**Reply:**

Thank you for noticing this, we will replace "times" by "3 h release time intervals".

---

## Author Response (AR2)

**Dear Editor,**

Please find below the revisions we did to comply with the Reviewers' comments:

\* Reviewer 2: Still, I suggest that the properties of the task (i.e. the CTBTO context with quite sparse measuring network; the assumption of knowledge of release rate boundaries of both, temporal and absolute/day) should be stated (defined) in the Introduction as the definition of your source reconstruction problem since it is quite specific.

**Reply:**

We have added to the introduction:

L 69: "In this study, the Bayesian source reconstruction tool FREARtool (FREAR stands for Forensic Radionuclide Event Analysis and Reconstruction) described in De Meutter and Hoffman (2020) will be used. FREAR was designed to determine the properties of a single point release (such as the release location, release amount and release start and release stop times) based on observations from one or more sparse measuring networks. Expert information can be taken into account through the prior distribution."

However, we disagree that our source reconstruction problem relies on "*the assumption of knowledge of release rate boundaries of both, temporal and absolute/day*)": knowledge of the release rate boundaries is not necessary to use the FREAR tool, since the tool can be run with wide priors. However, if expert information is available, it is clearly recommended to incorporate such information through the prior distribution.

\* *Reviewer 3: There are two more specific comments I have for this revision.*

*Line* 5, "The Bayesian approach has the advantage of providing credible intervals on the inferred source parameters in a natural way."

Although the authors explained in their responses to my previous comments, "credible intervals" is still too arbitrary to be meaningful. In addition, what can be defined as "a natural way" is questionable as well. This sentence is really too ambiguous to be included in the succinct abstract.

**Reply:**

We have rewritten the original sentence as follows:

L 5: "The Bayesian approach has the advantage of providing <del>credible intervals</del> an uncertainty quantification on the inferred source parameters <del>in a natural way</del>."

Line 167, "... a uniform prior between 10 and 16"

Based on what the authors stated in their responses to my previous comments ("the prior distribution from which initial samples are drawn is a uniform distribution between 10 and 16), it is mistaken to call it "a uniform prior". Please correct this.

**Reply:**

We have rewritten that sentence as:

L 150: "Since this spans many orders of magnitude, we take log 10(Q) as source parameter in our implementation and simply impose a uniform prior between 10 and 16 use a uniform distribution between 10 and 16 as uninformative prior."

Furthermore, we have added in the beginning of that paragraph:

L 146: "Uninformative bounded uniform priors are used for the source parameters. The prior is designed to allow for all plausible scenarios given the sparse measurement network and under the assumption the detected radionuclides are from the same release. For the current study, the source longitude is assumed to be between 20° and 80° and the source latitude is assumed to be between 40° and 70° (see Fig. 1 for a map showing the search domain)."

Yours faithfully, The authors